# Rapid evolution of an adaptive multicellular morphology of *Candida auris* during systemic infection

Jian Bing[1,2,10], Zhangyue Guan[1,10], Tianhong Zheng[1], Craig L. Ennis[3,4], Clarissa J. Nobile [3,5], Changbin Chen [6], Haiqing Chu [7,8] & Guanghua Huang [1,9]

*Candida auris* has become a serious threat to public health. The mechanisms of how this fungal pathogen adapts to the mammalian host are poorly understood. Here we report the rapid evolution of an adaptive *C. auris* multicellular aggregative morphology in the murine host during systemic infection. *C. auris* aggregative cells accumulate in the brain and exhibit obvious advantages over the single-celled yeast-form cells during systemic infection. Genetic mutations, specifically de novo point mutations in genes associated with cell division or budding processes, underlie the rapid evolution of this aggregative phenotype. Most mutated *C. auris* genes are associated with the regulation of cell wall integrity, cytokinesis, cytoskeletal properties, and cellular polarization. Moreover, the multicellular aggregates are notably more recalcitrant to the host antimicrobial peptides LL-37 and PACAP relative to the single-celled yeast-form cells. Overall, to survive in the host, *C. auris* can rapidly evolve a multicellular aggregative morphology via genetic mutations.

The emerging human fungal pathogen *Candida auris* was first described in 2009 in Japan and is becoming a serious global health threat[1–3]. Due to the notable increase in the incidence of *C. auris* infections worldwide, the rapid transmission and outbreak potential of *C. auris*, its multidrug resistance properties, and its persistence in the environment, the Centers for Disease Control and Prevention (CDC; https://www.cdc.gov) has issued several clinical alerts to healthcare facilities to be on the lookout for *C. auris* infections in patients and the World Health Organization (WHO; http://www.who.int) has added *C. auris* to the 2022 fungal priority pathogens list[4]. Like other pathogenic *Candida* species such as *Candida albicans* and *Candida tropicalis*, *C. auris* is

able to colonize the ears, wounds, respiratory tracts, urinary tracts, and skin of immunocompetent individuals, and can also cause serious bloodstream and/or invasive infections largely in immunocompromised patients[2,3].

Given its recent emergence and rapid spread worldwide, understanding the mechanisms of how *C. auris* rapidly evolved from a benign environmental fungus to a successful fungal pathogen is of utmost importance to controlling and preventing outbreaks of *C. auris* and other emerging fungal pathogens in the future. Morphological plasticity is a general and critical strategy used by pathogenic bacteria and fungi to adapt to changing environmental conditions and survive

[1]Shanghai Institute of Infectious Disease and Biosecurity, Department of infectious diseases, Huashan Hospital and State Key Laboratory of Genetic Engineering, School of Life Sciences, Fudan University, Shanghai 200438, China. [2]Shanghai Engineering Research Center of Industrial Microorganisms, Shanghai 200438, China. [3]Department of Molecular and Cell Biology, University of California, Merced, Merced, CA 95343, USA. [4]Quantitative and Systems Biology Graduate Program, University of California, Merced, Merced, CA 95343, USA. [5]Health Sciences Research Institute, University of California, Merced, Merced, CA 95343, USA. [6]The Center for Microbes, Development, and Health, Key Laboratory of Molecular Virology and Immunology, Unit of Pathogenic Fungal Infection & Host Immunity, Shanghai Institute of Immunity and Infection, Chinese Academy of Sciences, Shanghai 200031, China. [7]Department of Respiratory and Critical Care Medicine, Shanghai Pulmonary Hospital, School of Medicine, Tongji University, Shanghai 200433, China. [8]Shanghai Key Laboratory of Tuberculosis, Shanghai Pulmonary Hospital, School of Medicine, Tongji University, Shanghai 200433, China. [9]College of Pharmaceutical Sciences, Southwest University, Chongqing 400716, China. [10]These authors contributed equally: Jian Bing, Zhangyue Guan. e-mail: chu_haiqing@126.com; huanggh@fudan.edu.cn

in the host[5–7]. For example, *C. albicans* can switch between multiple morphological phenotypes in response to in vivo and in vitro environmental changes[8]. White-opaque switching and yeast-filament transitioning are among the most well-studied morphological transitions investigated in *C. albicans* over the past two decades[8–10]. Different morphological cell types of *C. albicans* often have distinct characteristics in term of host invasion, stress resistance, tissue tropism, and susceptibility to antifungal drugs and the host immune system[8–10]. For example, in *C. albicans*, multicellular filaments are more invasive than yeast-form cells, while white cells are more virulent than opaque cells in mouse models of systemic infections[10].

Numerous environmental factors and genetic regulators and pathways are involved in the control of morphological transitions in *C. albicans*[8,10–12]. For example, increased ambient temperatures and $CO_2$ levels, serum, neutral or basic pH, and N-acetylglucosamine promote the growth of filaments, while decreased ambient temperatures, acidic pH, and rich nutrient conditions induce the transition from the filamentous to the yeast form[11]. The cAMP/PKA signaling and the Ste11-Hst7-Cek1/2 MAPK pathways function as major regulators in controlling filamentation in *C. albicans*[11,13,14]. Activation of these two conserved pathways leads to changes in expression of downstream transcription factors and ultimately the development of filaments. Moreover, genes involved in the regulation of cell wall integrity, formation of septin rings, cytokinesis, and cytoskeletal polarization also often play important roles in mediating morphological transitions in *C. albicans*[11,15,16]. Environmental cues induce morphological transitions in pathogenic *Candida* species largely through the regulation of the aforementioned signaling pathways by epigenetic or non-genetic changes[11,12,15,16]. Recent studies have additionally shown that genetic mutations are also involved in shaping the virulence or commensal characteristics of certain *Candida* species[17–19].

In response to host environmental stresses, pathogenic *Candida* species often undergo microevolution driven by genetic mutations to adapt to the host during commensal colonization or infection[18–23]. For example, serial passage of *C. albicans* through the mouse gut induced the adaptive evolution of a low-virulence phenotype with an enhance ability to colonize the mammalian gut[18]. Passaging *C. albicans* in this way led to mutations in several genes required for *C. albicans* filamentous growth, including *FLO8* and *EFG1*. In another example, continuous coincubation of *Candida glabrata* with host murine macrophages led to the formation of a pseudohyphal-like morphology and a mutation in the chitin synthase-encoding *CHS2* gene[22]. This evolved pseudohyphal-like form allowed *C. glabrata* to rapidly escape from macrophages and the pseudohyphal-like cells displayed increased virulence during host infection. The soil fungal pathogen *Cryptococcus neoformans* is also capable of undergoing microevolution to survive in the host. Coincubation of *C. neoformans* with its environmental amoeba predator has been shown to induce a pseudohyphal form of *C. neoformans* caused by DNA mutations in genes encoding components of the Regulation of Ace2 Morphogenesis (RAM) pathway[23]. Taken together, numerous pathogenic fungi undergo modifications of their genomic sequences to enhance their abilities to adapt to changing environmental conditions that occur during host colonization and infection. These small-scale genetic changes may be efficient drivers of microevolution in these species.

Like *C. albicans*, *C. auris* also has several cellular morphologies including yeast and filamentous forms[2]. It has been demonstrated that morphological transitions are a general feature of clinical *C. auris* strains of different genetic clades[24]. A number of transcriptional and cell wall regulators are involved in the regulation of filamentous growth and aggregation in *C. auris*[25]. Genotoxic stress and inactivation of the molecular chaperone Hsp90 also promote the growth of filaments in *C. auris*[26,27], indicating that its ability to undergo morphological transitions has conserved regulatory features. One study demonstrated that DINOR, a long non-coding RNA, plays a critical role

in the control of filamentation and virulence in *C. auris*[28]. Deletion of DINOR resulted in DNA damage and filamentous growth, suggesting that genetic alterations could be involved in the regulation of morphogenesis in *C. auris*. Interestingly, clinical isolates of *C. auris* rarely undergo filamentation, although a subset of *C. auris* cells can become filament-competent (acquiring the ability to develop filaments) after passage through the mammalian host[29]. The mechanism underlying this host-induced filamentation in *C. auris* remains to be investigated. Since the filamentous and filament-competent cells exhibit cellular memory of their cell state, we suspect that genetic or epigenetic changes could be involved in this regulation[2].

Another striking feature of *C. auris* is its ability to form multicellular aggregates[25,30–32]. Numerous clinical isolates of *C. auris* can form aggregative cells that differ from single-celled yeast-form cells in a number of biological aspects including colony and cellular morphologies, biofilm-forming abilities, susceptibilities to antifungal drugs, susceptibilities to immune cells, and virulence[24,25,30–33]. The aggregative form could be induced under both in vivo and in vitro conditions[24,25,30–32]. For example, exposure to the antifungal drug caspofungin, an inhibitor of β−1,3-glucan synthase, caused not only modifications of the cell wall but also led to morphological changes and the formation of aggregative cells in *C. auris*[34]. Forgacs et al. [35] observed the presence of large aggregates of *C. auris* cells in the heart, kidney, and liver tissues in a neutropenic mouse model, suggesting that numerous clinical isolates are capable of forming aggregates in the host[35]. We and others recently found that there are two types of aggregative morphologies in *C. auris*[30,36,37]. One type is formed due to a defect in cell division, while the other type is thought to be the result of the overexpression of genes associated with adhesion. These two aggregative morphologies of *C. auris* differ in their biofilm forming abilities and virulence[30]. The host environmental inducers, biological bases, and regulatory mechanisms of the former aggregative phenotype, however, remain unknown.

In this study, we report that a subset of *C. auris* cells can rapidly evolve from the single-celled yeast form to a multicellular aggregative morphology during systemic host infection in a mouse model. Scanning electron microscopy (SEM) and transmission electron microscopy (TEM) analyses indicate that formation of this aggregative phenotype was caused by cell division or budding defects. Genetic mutations in genes associated with the regulation of cell wall integrity, cytokinesis, cytoskeletal properties, and cellular polarization were responsible for the rapid evolution of this multicellular aggregative phenotype. Compared to the yeast-form morphology, these *C. auris* aggregative cells exhibited a general increased ability to survive in the host and an increased capacity to colonize the brain. Moreover, aggregative cells were more resistant to phagocytosis by host macrophages and host-derived antimicrobial peptides. This rapidly evolved aggregative morphology likely represents an adaptive strategy of *C. auris* in response to the host environment and attack by the host immune system.

## Results

### Aggregate formation in C. auris can be induced during systemic infection in the host

A unique characteristic of *C. auris* is that some clinical isolates exhibit a multicellular aggregative phenotype[31–33,38]. As shown in Fig. 1 and in previous studies[30–33], this multicellular morphology is associated with defects in cell division. The *C. auris* mother and daughter cells within the multicellular aggregates remain attached caused by an inability to release daughter cells after budding. In our previous studies, we observed that rough or wrinkled colonies of *C. auris* occasionally contained filamentous or aggregative cells when *C. auris* was recovered from mouse tissues during systemic infection assays[24,29]. To reveal whether the mammalian host is able to induce the formation of *C. auris* aggregates, we revisited these mouse systemic infection experiments in

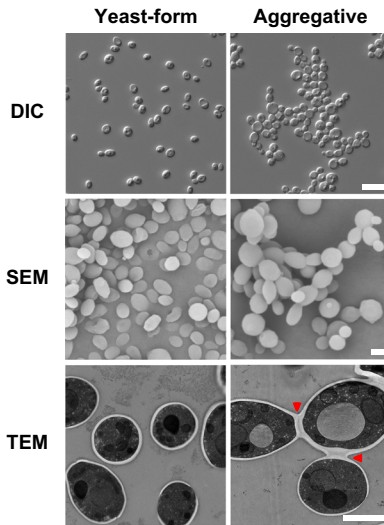

**Fig. 1 | Yeast-form and aggregative morphologies of a clinical isolate of *C. auris*.** Strain used: AR0386 and AR0386A. *C. auris* cells were plated onto YPD plates at 30 °C for 4 days. DIC differential interference contrast, SEM scanning electron microscopy, TEM transmission electron microscopy. Red arrows indicate inter-cellular septa. Scale bar for DIC, 10 μm; for SEM and TEM, 2 μm.

more detail (Fig. 2a). *C. auris* yeast-form cells of 11 clinical strains spanning the four major *C. auris* genetic clades were injected into the mice via the tail vein (Table S1). After 3 days of infection, we recovered fungal cells from the brain, liver, spleen, lung, and kidney and replated them onto YPD medium containing the red dye phloxine B. This dye was used in the screening assay because it largely stains colonies containing aggregative cells of *C. auris* pink or red and facilitated our abilities to identify evolved mutant strains. The average frequency of aggregative mutant strains from brain tissue was 0.011%, whereas the average frequency from the liver, spleen, lung, and kidney tissues were 0.0019%-0.0086%. Of note, the fungal burden in the blood was extremely low and we did not obtain aggregative mutants from the bloodstream. As shown in Fig. 2b, in total, we identified 96 rough or pink/red colonies that contained aggregative cells of *C. auris* from the mice infected with 8 strains (8/11, 72.72%). Of them, the majority (54/96, 56.25%) were isolated from the brain, implying that aggregative cells have a tissue tropism for the mouse brain during systemic infections. The morphologies of the aggregative cells at the cellular level were examined using differential interference contrast (DIC) microscopy, SEM, and TEM assays. Several representative cellular morphologies are shown in Figs. 2c and S1. Our results indicate that the mammalian host provides a selective advantage for cells capable of generating the multicellular aggregative morphology during systemic infections.

To exclude the possibility of potential bias being introduced in our mutation screen due to the use of the red dye phloxine B, we performed comparative experiments using media with and without phloxine B. Brain tissues from 12 *C. auris* infected mice were broken up and equally plated onto YPD plates with and without phloxine B. We identified 11 evolved aggregative mutant strains from the medium containing phloxine B and 6 mutant strains from the medium lacking phloxine B (Table S2 and Dataset S1). Therefore, the addition of phloxine B facilitated our abilities to identify evolved mutant strains and did not introduce bias in our mutation screen.

## Genetic mutations underlie the rapid evolution of the *C. auris* multicellular aggregative morphology

We observed that *C. auris* cells within all 113 identified colonies with a rough or red appearance exhibited heritability for the aggregative phenotype under in vitro culture conditions, suggesting that genetic or epigenetic alterations could be at play in the regulation of this multicellular aggregative morphology. To uncover the underlying regulatory mechanism, we performed next generation whole-genome sequencing (WGS) with all evolved aggregative strains. Single-nucleotide polymorphism (SNP) and copy number variation (CNV) analyses were performed, and the detailed results are presented in Dataset S1. To our surprise, all aggregative strains isolated from the mouse host carried one or more mutations or copy number losses in the open reading frame (ORF) regions of a set of genes associated with cell division and/or budding. In total, 48 distinct mutations were identified, which involved 31 independent genes (Fig. 3a and Dataset S1). We observed missense, nonsense, frameshift, and gene loss mutations within these regions (Dataset S1). Functional category analysis indicated that the major of mutated *C. auris* genes in these regions were enriched for genes encoding proteins involved in the regulation of cell wall integrity, cytokinesis, cytoskeletal properties, and/or cellular polarity (Figs. 3 and S2). Notably, mutations within the chitin synthase-encoding gene *CHS1* were found in 39 evolved aggregative *C. auris* strains and included four distinct types of mutations (Dataset S1). Chs1 is required for primary septum synthesis and cell wall integrity in *C. albicans*[39]. Similar to its *C. albicans* counterpart, *CHS1* in *C. auris* encodes a class II chitin synthase. Mutation of *CHS1* led to a potential aberrant septa in *C. auris* cells (Fig. 2c). Other notable mutated *C. auris* genes included *KIN3* (14 isolates), *ACE2* (8 isolates), *CAS4* (6 isolates), *LRG1* (5 isolates), *APM1* (4 isolates), *BNI1* (4 isolates), *KIC1* (3 isolates), and *LAA1* (3 isolates), which are all involved in cytokinesis, cytoskeletal properties, and/or cellular polarity processes. A summary of the associated signaling pathways and cellular processes for the identified mutated genes is presented in Fig. 3b.

We note that several of the mutated *C. auris* genes (*CBK1*, *MOB2*, *HYM1*, *KIC1*, *ACE2*, and *CAS4*) identified in the *C. auris* evolved strains (Fig. 3) are related to fungal genes that are members of the well-conserved Regulation of Ace2 Morphogenesis (RAM) pathway, which regulates cellular polarity, cell separation, and filamentation in *C. glabrata*, *C. albicans*, and *Saccharomyces cerevisiae*[40–43]. The Ace2 transcription factor regulates chitinase (Cht1) that is required for septal plate dissolution. Consistently, Santana and O'Meara demonstrated that inactivation of Ace2 caused a cell separation defect in *C. auris*[25]. Tian et al.[44] recently isolated two aggregative *C. auris* clinical strains carrying *ACE2* mutations[44]. Given their conserved roles in cell growth or the cell cycle in other fungi, it is reasonable that mutations in genes associated with these pathways could lead to defects in budding and/or cell division in *C. auris*.

Four genes (*LRG1*, *BNI1*, *IQG1*, and *HOF1*) associated with the regulation of cytokinesis were identified in the evolved *C. auris* aggregative strains (Fig. 3). Lrg1 is a GTPase-activating protein (GAP) that regulates Cdc42 and Ras1, which control morphogenesis through multiple signaling pathways in fungi[45,46]. The formin protein Bni1, IQGAP protein Iqg1, and F-BAR protein Hof1 are downstream of Lrg1. Lrg1 and Bni1 regulate cellular polarity, while Hof1 is required for assembly and contraction of the actomyosin ring (AMR)[45–47]. We note that several genes identified in the mutated list are involved in the regulation of multiple biological processes, such as *LRG1* and *BNI1* (Fig. 3b, c).

In addition to *LRG1* and *BNI1*, mutations in two other genes (*HSL1* and *KIN3*) were also identified in the in vivo evolved strains. Hsl1 and Kin3 play critical roles in cellular polarity and budding in several yeast species[48]. Bni1 are core components of the polarisome[49], while Hsl1 and Kin3 are kinases involved in the regulation of septin assembly and filamentation[50,51]. Mutations in seven genes Pan1, Arc19 and AP adaptors (Apl2, Apl4, Laa1, Apm1, and Bsp1), which are core proteins involved in cytoskeletal organization, were also identified (Fig. 3). Moreover, we identified 10 genes involved in the regulation of cell wall integrity, including *DIG2* (encoding a MAPK proteins), *LRG1*, *BNI1* and several genes associated with cell wall formation (*CHS1*, *MNN11*, *KRE5*, *KTR2*, *OCH1*, *VAN1*, and *VRG4*). Of these genes, three (*OCH1*, *MNN11*,

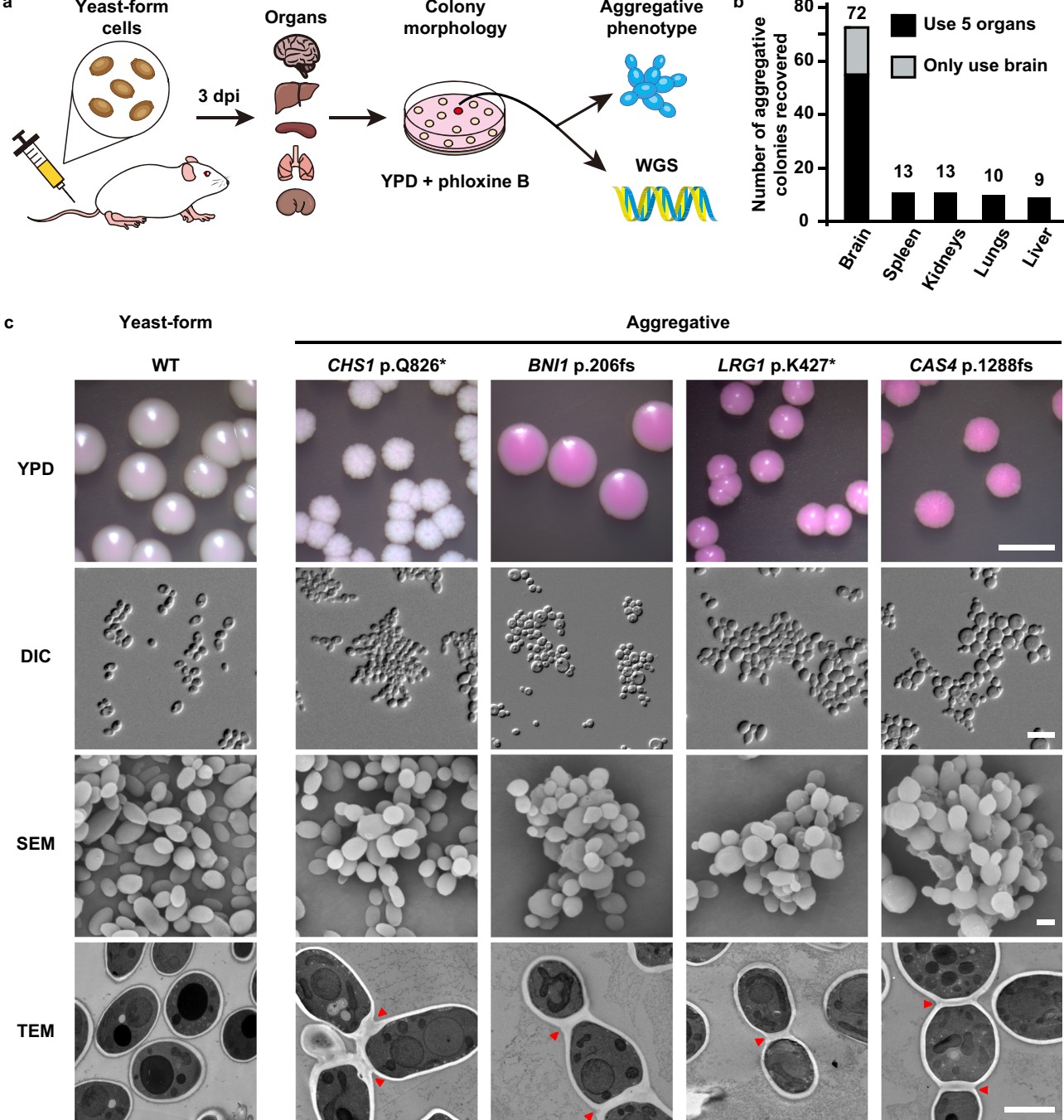

**Fig. 2 | Experimental evolution of a typical yeast-form strain of *C. auris* into a multicellular aggregative morphology. a** Schematic of aggregative colony acquisition from a mouse systemic infection system. Six-week-old mice were injected with the yeast-form cells of *C. auris* via the tail vein. Fungal cells were recovered from the brain, liver, spleen, lung, and kidney tissues after 3 days post infection (dpi) and plated onto YPD medium plates containing the red dye phloxine B. Pink, rough or wrinkled colonies containing aggregative cells were subject to microscopy assays and whole genome sequencing (WGS). **b** Number of evolved aggregative *C. auris* strains isolated from different mouse organs. In total, 113 aggregative mutant strains were obtained. Of them, 96 strains (black rectangles) were isolated from the brain, liver, spleen, lung, and kidney tissues of 28 mice and 17 strains (gray rectangle) were isolated from the brain tissue of 12 mice. Detailed strain information is presented in Tables S1, S2, and Dataset S1. **c** Colony and cellular morphologies of the WT (BJCA001) and four representative aggregative strains evolved during mouse systemic infections. *CHS1* p.Q826* (FDAG4), *BNI1* p.206 fs (FDAG30), *LRG1* p.K427* (FDAG9), and *CAS4* p.1288 fs (FDAG1) represent the four evolved strains with mutations in *CHS1, BNI1, LRG1,* or *CAS4,* respectively. *C. auris* cells were plated onto YPD plates supplemented with phloxine B at 30 °C for 4 days. *, nonsense mutations; fs, frameshift mutations. Red arrows indicate intercellular septa. Scale bar for colony, 5 mm; for DIC morphology, 10 μm; for SEM and TEM morphologies, 2 μm.

and *VAN1*) are required for the initiation and elongation of N-mannosylation of cell wall in yeasts[52]. Gene ontology category analysis further confirmed the genetic and functional associations we observed among the identified mutated genes (Fig. S2). Moreover, we found that evolved aggregative mutants of the same genes that derived from different parental strains (of different genetic clades) exhibited a similar phenotype (Fig. S3), indicating the general characteristic of this phenomenon in clinical strains.

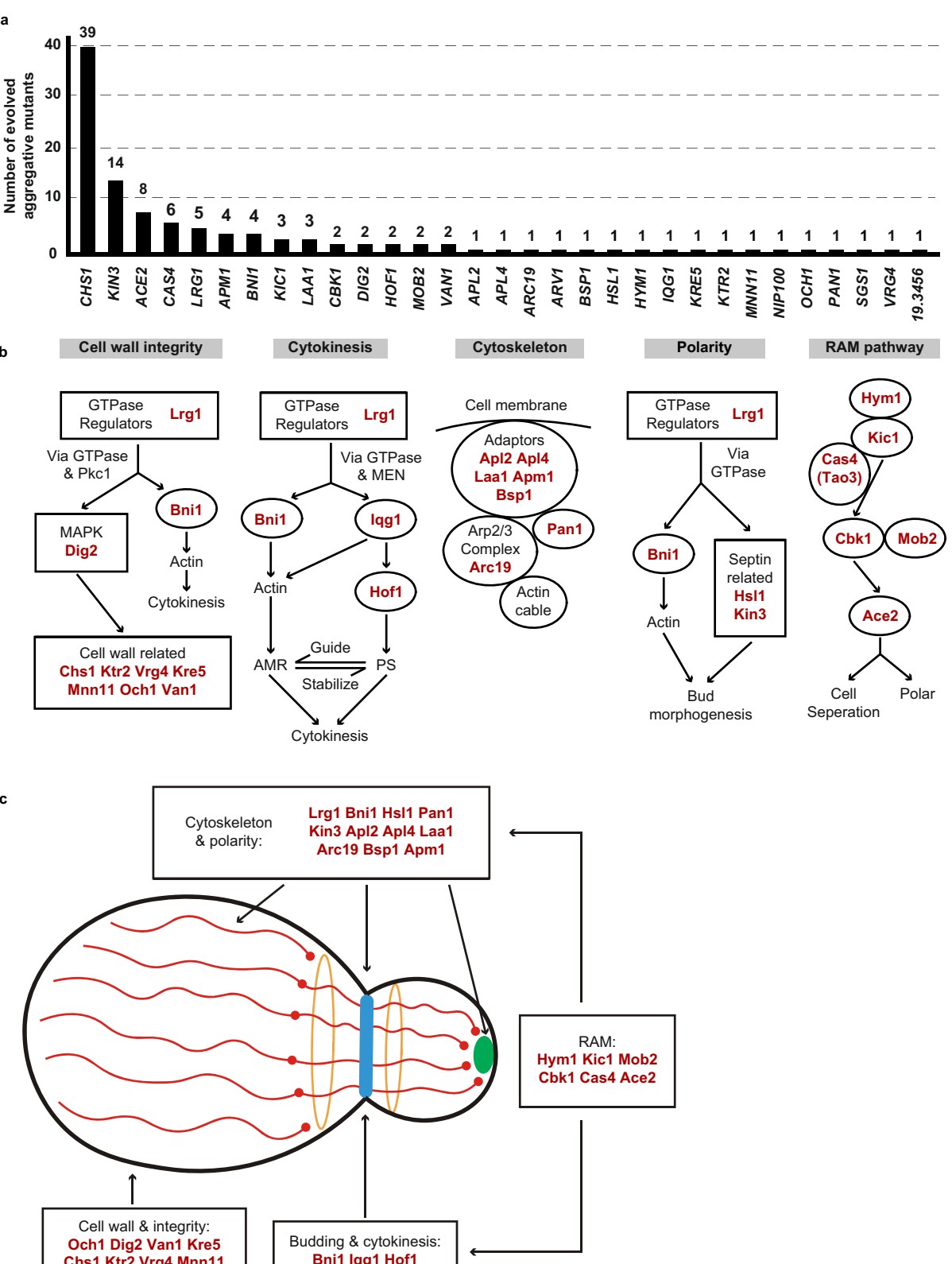

A mutation in *SSD1* (p.R549K) was found in the aggregative cells of the *C. auris* clinical isolate AR0386A. *SSD1* encodes a translational repressor and is involved in polar growth in *S. cerevisiae*[53,54]. This mutation was located within a region of *SSD1* that encodes the conserved domain III of the protein, which could attenuate or inactivate Ssd1 function and thus lead to the observed aggregative phenotype (Fig. 1). A similar aggregative phenotype was observed in the evolved strain carrying the same site mutation we generated in the control strain (Fig. S4). These findings demonstrate that the p.R549K mutation in *SSD1* is directly related to the aggregative phenotype in *C. auris*.

To reveal whether mutations in cell budding- or cell division-associated genes are prevalent in *C. auris* clinical strains, we next analyzed 4482 genomic sequences of *C. auris* available in the NCBI database. As shown in Fig. S5 and Dataset S2, based on the mutated

**Fig. 3 | Mutated genes of evolved aggregative strains are involved in the regulation of cell division, cytoskeletal properties and cellular polarity.** These genes were independently identified from different evolved aggregative mutant strains. **a** Number of isolates for each gene mutation. All aggregative mutant strains were recovered from mouse organs initially infected with yeast-form *C. auris* cells (as shown in Fig. 2). The mutated loci were identified by comparing the genomic sequences of the mutant strains with those of the yeast-form parent strain. **b** Major biological processes or signaling pathways of the mutated genes (highlighted in red) of the evolved aggregative strains. Cell wall integrity pathway: 14 mutations were identified, including mutations in genes encoding chitin synthase Chs1, GTPase regulator Lrg1, MAPK proteins Dig2, and several cell wall proteins. Cytokinesis: 7 mutations were identified, including mutations in *LRG1* and genes encoding the formin protein Bni1, F-BAR protein Hof1, and IQGAP protein. Actin cable and cytoskeleton: 8 mutations were identified, including mutations in genes encoding components of the actin cytoskeleton-regulatory complex Pan1, clathrin

and adaptor complex (Apl2, Apl4, Apm1), and the Arp2/3 complex (Arc19). Cellular polarity: 6 mutations were identified, including mutations in *LRG1*, *BNI1*, and genes encoding kinases Kin3 and Hsl1 associated with septin formation. RAM pathway: 13 mutations were identified, including mutations in *ACE2*, *CBK1*, *MOB2*, *HYM1*, *KIC1*, and *CAS4*. **c** Schematic model indicates the cellular functions of the identified mutated genes. Most mutated genes are involved in the regulation of cell budding and/or cell division. The blue strip indicates the septum; red lines indicate actin cables involved in exocytosis; red circles indicate actin patches involved in endocytosis; orange circles indicate septin double rings in endocytosis; the green solid circle indicates protein Bni1; and the peripheral shadow layer indicates the cell wall. Mutated genes identified in this study are highlighted in red. The mutated genes/pathways identified in this study are interconnected and several genes are involved in the regulation of multiple processes. Note that the biological processes or signaling pathways of the *C. auris* mutated genes shown were predicted based on homology to their *S. cerevisiae* or *C. albicans* counterparts (**b**) and (**c**).

genes identified in our evolved aggregative strains, we identified 11 strains containing gene mutations associated with cell budding or cell division processes. Specifically, 9 types of mutations (*ACE2* Q163*, *BNI1* p.T484 fs, *CAS4* p.E1289 fs, *CAS4* p.Q2657*, *ENO1* p.K270 fs, *INPS2* p.E921*, *LRG1* p.V434 fs, *LRG1* p.S296 fs, *PAN1* p.E1047 fs) were identified (only nonsense and frameshift mutations were considered), several of which overlapped with those identified in the evolved aggregative strains (Fig. 3). These findings indicate that mutations in cell budding- or cell division-associated regulators are present in *C. auris* clinical strains.

Taken together, most mutations identified in the *C. auris* evolved aggregative isolates likely impact biological processes involved in regulating cell budding or cell division, such as cytokinesis, cytoskeletal properties, cellular polarity, and cell wall integrity (Fig. 3b, c). Most of the genes identified in our mutational screens are conserved in *C. albicans* and *S. cerevisiae* and play critical roles in the regulation of cell aggregation, and filamentous or invasive growth in these yeast species (Table S3). It is reasonable to predict that genetic perturbations of these processes would lead to the evolution of the multicellular aggregative morphology, which could benefit *C. auris* by allowing it to adapt to the changing host microenvironment and thus persist and survive in the mammalian host.

## Verification of the association between mutated genes and the aggregative phenotype of the evolved strains

To test whether inactivation of the aforementioned major genes identified in the evolved strains is sufficient to cause the aggregative phenotype, we constructed a set of deletion or substitution mutant strains of 14 representative genes (*ace2Δ, apm1Δ, apl2Δ, bni1Δ, cas4Δ, chs1Δ, hof1Δ, iqg1Δ, irg1Δ, och1Δ, nip100Δ, sgs1Δ, ssd1Δ*, and *van1Δ*) in a *his1* deletion strain of BJCA001. These genes play central roles in the regulation of cytokinesis, cytoskeletal properties, cellular polarity, and/or cell wall integrity in fungal species. As shown in Fig. S4, all mutant strains constructed showed similar cellular and colony morphologies to those of their corresponding evolved aggregative strain counterparts. The mutant strains also exhibited cell budding and/or cell division defects, suggesting that the identified host-induced genetic mutations of the evolved strains were sufficient to cause the generation of the aggregative morphology in *C. auris*.

## Global transcriptional expression profiles of the yeast-form and aggregative isolates of *C. auris*

To further explore the mechanism of formation of the aggregative phenotype, we performed transcriptomic analysis using RNA-Seq assays in the yeast-form strain (BJCA001) and four aggregative isolates (*CHS1* p.Q826*, *BNI1* p.206 fs, *LRG1* p.K427*, and *CAS4* p.1288 fs mutants). Three biological repeats were performed. Principal Component Analysis (PCA) analysis indicated the consistency of expression profiles among the repeats (Fig. S6). As shown in

supplementary Dataset S3, 62, 649, 833, and 1994 differentially expressed genes (DEGs) were identified between the WT (yeast-form) and *CHS1* p.Q826* mutant, WT and *BNI1* p.206 fs, WT and *LRG1* p.K427*, and WT and *CAS4* p.1288 fs mutants, respectively. Hierarchical clustering analysis of DEGs indicated a high transcriptional expression similarity between the yeast-form strain and *CHS1* p.Q826* mutant and also between the *BNI1* p.206 fs and *LRG1* p.K427* mutants (Fig. 4a). There were 445 common DEGs between the *BNI1* p.206 fs and *LRG1* p.K427* mutants (Fig. 4b, c). Many of them are involved in the regulation of cytoskeleton, cell cycle or cytokinesis, and cell wall integrity (CWI) (Fig. 4b and Dataset S3). These results are reasonable because Bni1 and Lrg1 are associated regulators involved in the regulation of the CWI[55], cytokinesis, and polarity pathways, while Cas4 is a member of the RAM pathway that regulates cell polarity and separation[42]. Mutation of *CAS4* affected the expression of nearly 2000 genes perhaps due to its global regulatory role in *C. albicans*. Mutation of *CHS1* only caused 62 DEGs perhaps due to the chitin synthase Chs1 representing a downstream component of the CWI pathway[56]. Consistent with the aggregative phenotype, we found a set of common DEGs associated with cytoskeleton, cell division, and cell wall integrity in all the four evolved mutants (compared to that in the yeast-form strain BJCA001, Fig. 4d and Dataset S3). Of them, *CDC5*, *CDC14*, *CDC20*, *ESC4*, and *HGC1* are involved in the regulation of cell division. *HGC1* encodes a G1 cyclin-related protein that is associated with the cyclin-dependent kinase Cdc28 and is essential for filamentous growth in *C. albicans*[15].

Compared to the yeast-form strain, 398 common DEGs were identified in the *BNI1* p.206 fs, *LRG1* p.K427*, and *CAS4* p.1288 fs mutants (Fig. 4e). Of these DEGs, a large subset of genes are associated with the regulation of cell wall integrity (e.g., *PKC1*, *BMT1*, *CHS1*, and *PGA6*) and some are regulators of cell budding (such as *IQG1* and *GIN4*). Moreover, many DEGs are associated basic metabolic processes (such as amino acid metabolism and protein synthesis), suggesting that mutation of these genes has a great impact on cell physiology. A large set of genes involved in the regulation of cell wall integrity, cell cycle, and cytoskeleton (e.g., *IQG1*, *OCH1*, *CHS1*, *LMO1*, *ENO1*, *CAS4*, *CHS3*, and *CHS6*) exhibited a similar expression pattern in the *BNI1* p.206 fs, *LRG1* p.K427*, and *CAS4* p.1288 fs mutants (Fig. 4e, f). In contrast, mutation of *CHS1* had a relatively weak impact on the general cell physiology. To further uncover the genetic and functional connections among the DEGs between the yeast-form strain and aggregative mutants (common DEGs found in two or more mutants), we performed network analysis using the STRING database[57]. Three major clusters involving 96 proteins with predicted protein-protein interaction relationship were identified by K-mean method. As shown in Fig. S7, cluster 1 consisted of 7 genes regulating the aggregative phenotype identified in the evolved strains in this study and some key regulators of cytokinesis and the CWI pathway. Cluster 2 proteins were mainly associated

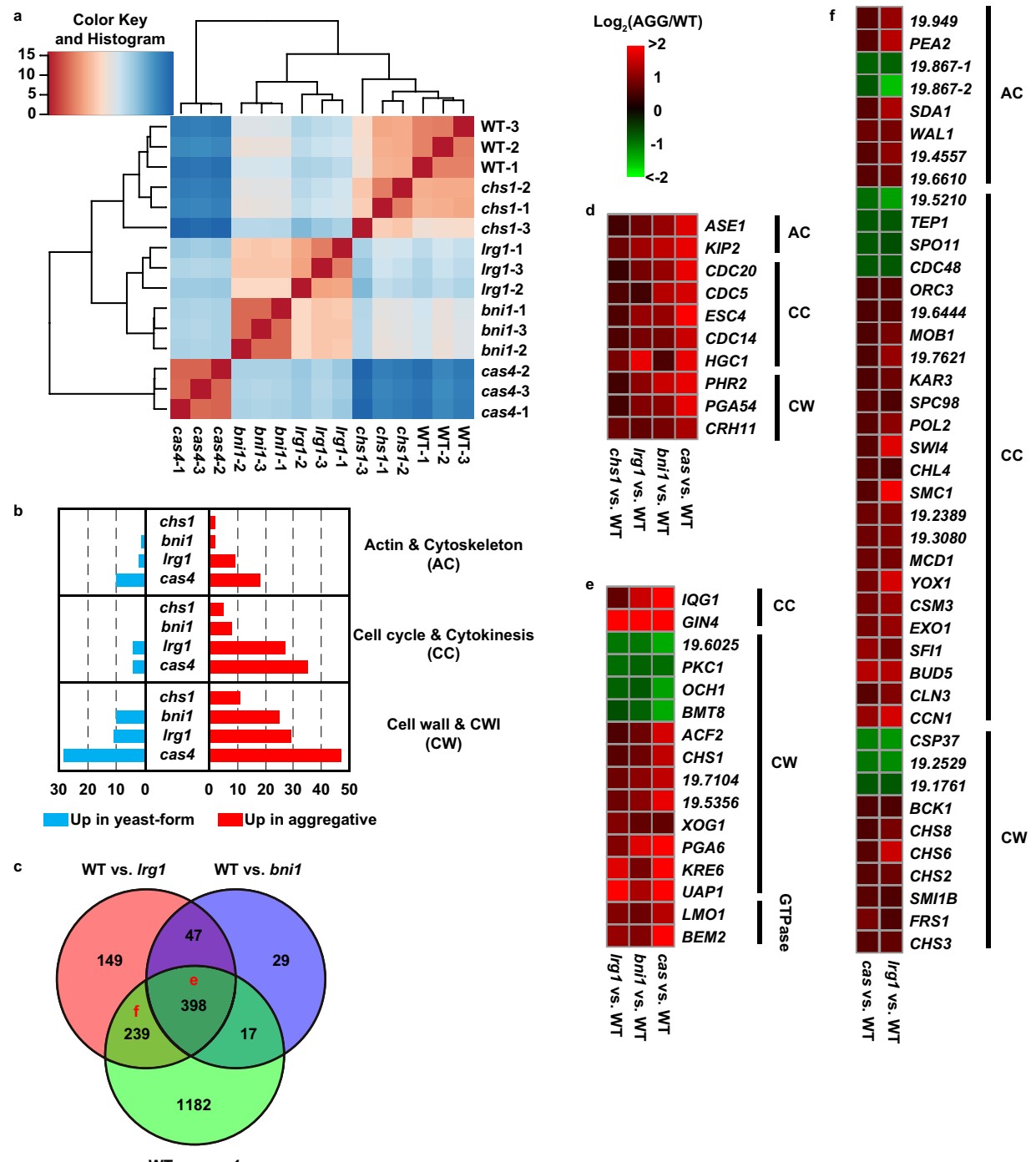

**Fig. 4 | Analysis of differentially expressed genes among the yeast-form strain and evolved aggregative isolates of *C. auris*.** Strains analyzed: BJCA001 (yeast-form), *CHS1* p.Q826* (*chs1*, FDAG4), *BNI1* p.206 fs (*bni1*, FDAG30), *LRG1* p.K427* (*lrg1*, FDAG9), and *CAS4* p.1288 fs (*cas4*, FDAG1). **a** Hierarchical clustering of differentially expressed genes (DEGs) between different strains using Euclidean distance. **b** Numbers of DEGs associated with CWI, cytoskeleton, and cell cycle/cytokinesis regulation between the yeast-form strain and *CHS1* p.Q826*, *BNI1* p.206 fs, *LRG1* p.K427*, or *CAS4* p.1288 fs mutants. **c** Venn plot of DEGs between the yeast-form strain and *BNI1* p.206 fs, *LRG1* p.K427*, or *CAS4* p.1288 fs mutants. **d** Representative common DEGs among the *CHS1* p.Q826*, *BNI1* p.206 fs, *LRG1* p.K427*, and *CAS4* p.1288 fs mutants. **e** Representative common DEGs among the *BNI1* p.206 fs, *LRG1* p.K427*, and *CAS4* p.1288 fs mutants. **f** Representative common DEGs between the *LRG1* p.K427* and *CAS4* p.1288 fs mutants. Log2 (fold change) values of DEGs between the yeast-form strain and corresponding mutant strains are shown in (**d**)−(**f**).

with cell cycle and cytokinesis. Cluster 3 proteins were majorly related to the cell wall integrity. These findings confirmed that the mutated genes of the aggregative isolates play critical roles in the regulation of CWI, cytoskeleton, cytokinesis, and cell polarity. Genetic perturbation of these pathways would lead to cell budding or division defects and formation of multicellular aggregates.

## Aggregative cells display an increased fitness and outcompete single-celled yeast-form cells in the brain during system infection

The ability to undergo morphological transitions is a common strategy used by pathogenic fungi to survive in the host and evade host immune attacks[11,58,59]. To compare the fitness and virulence of aggregative and

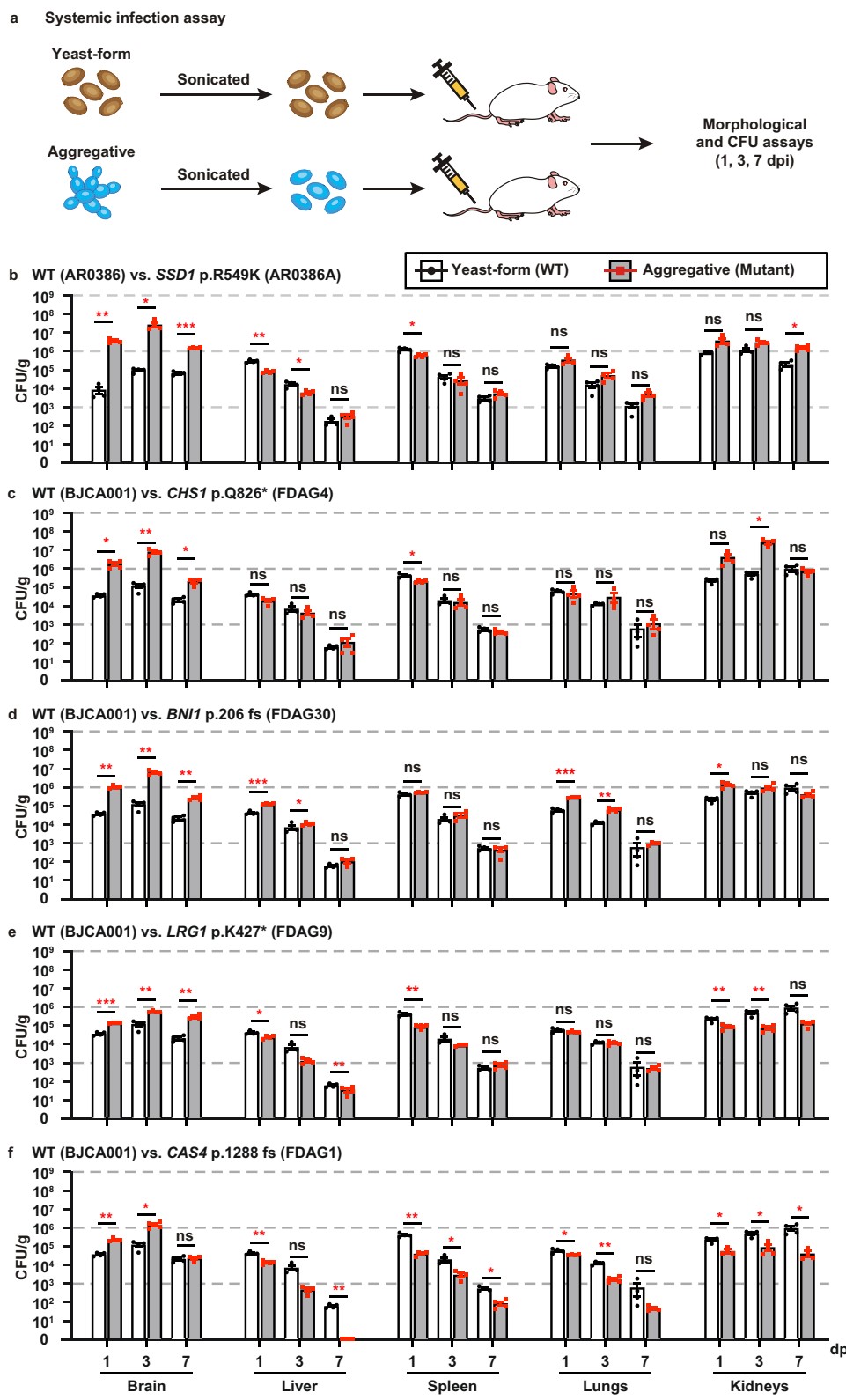

single-celled yeast-form cells of *C. auris*, we performed systemic infection and fungal burden assays using a mouse infection model using a selection of WT yeast-form and evolved mutant aggregative strains. To avoid blood vessel blockage and rapid animal death caused by *C. auris* aggregates, we converted *C. auris* aggregative cells to single cells by sonication prior to tail veil injection in the mice (Fig. 5a). *C. auris* cells were recovered from the brain, liver, spleen, lung, and kidney, and fungal burdens were calculated at 1, 3, and 7 days post infection (dpi).

A comparison of fungal burdens of the WT yeast-form to those of the evolved mutant aggregative strains is presented in Figs. 5b–f and S8a–e. Generally, the fungal burdens in the brain and kidney were much higher than those in the liver, spleen, and lung for both cell types of the WT yeast-form and the evolved mutant strains. This observation

**Fig. 5 | Comparative analyses of fitness and virulence of the yeast-form and aggregative cells using a competing mouse systemic infection model.**
**a** Schematic of the systemic infection assay. To convert to separated single cells for systemic infections, *C. auris* aggregative cells were subjected to sonication. To control for any potential negative effects of sonication, yeast-form cells were also treated with the same sonication protocol. Fungal cells ($1 \times 10^7$) were injected into each mouse via the tail vein. At 1, 3, or 7 dpi, fungal cells were recovered from the brain, liver, spleen, lung, and kidney tissues, weighed, and pulverized using Zirconia beads (3.2 mm) with 60 Hz power for 120 s. The homogenized samples were then re-plated onto YPD medium plates containing the red dye phloxine B. CFU assays (per gram organ) were performed. **b–f** Comparison of fungal burdens of the WT and evolved mutant strains in different mouse organs (4 mice per strain were used for each time point). **b** WT (yeast-form, AR0386) versus *SSD1* p.R549K (AR0386A). Strains AR0386 and AR0386A were isolated from the same patient. **c–f** WT (yeast-form, BJCA001) versus *CHS1* p.Q826* (FDAG4), *BNI1* p.206 fs (FDAG30), *LRG1* p.K427* (FDAG9), and *CAS4* p.1288 fs (FDAG1). The *P* value was determined by two-tailed paired Student's *t*-tests. *$P < 0.05$; **$P < 0.01$; ***$P < 0.001$. Data shown represents mean ± SD.

was more apparent after 3 dpi. Compared to the corresponding WT yeast-form control, the evolved aggregative mutant strains of *SSD1*, *CHS1*, *BNI1*, *LRG1*, *CAS4* (Fig. 5b–f), *KIN3*, *APL2*, *VAN1*, *NIP100*, and *ACE2* (Fig. S8a–e) exhibited remarkably increased fungal burdens in the brain (approximately 5 to 2000 fold-increases for the evolved aggregative mutant strains after 1 or 3 dpi). Evolved mutant aggregative strains of *LRG1*, *CAS4*, *BNI1*, *APL2*, *NIP100*, and *ACE2* also displayed increased fungal burdens in the kidney for at least two of the timepoints tested. We note that all mice infected with the evolved aggregative mutant strain of *ACE2* displayed rapid weight loss and died at 5 dpi. The fitness of the different evolved aggregative mutant strains in the liver, spleen and lung varied dramatically in terms of their fungal burdens and/or infection times (Fig. 5b–f and S8a–e).

To further explore the differences caused by the yeast-form and aggregative cells of *C. auris* during systemic infection in the host, we performed periodic acid-Schiff (PAS) staining and immunohistochemical analyses on the brain tissues. The WT yeast-form strain and two representative evolved aggregative mutant strains (*CHS1* p.Q826* and *BNI1* p.206 fs) were analyzed. As shown in Fig. S9a, obvious tissue damage and many dark spots indicative of tissue necrosis were observed in the brain tissues infected with the aggregative mutant strain cells but not in those infected with the WT yeast-form strain cells. Pathological and immunohistochemical analyses showed an increase in the number of monocytes and neutrophils within the tissues colonized with the aggregative cells (Fig. S9b). These results demonstrate that aggregative cells of *C. auris* can survive better in the murine host than yeast-form cells and have a clear tropism for the brain.

To further demonstrate the increased survivability of the *C. auris* aggregative cells compared to the WT yeast-form cells, we performed pairwise competition infection assays. A 50:50 mixture of the WT yeast-form strain cells and evolved aggregative mutant strain cells was injected into the mouse via the tail vein (Fig. 6a). After 1, 3, and 7 dpi, fungal burdens for each cell type in the brain, kidney, liver, spleen, and lung were examined. Yeast-form cells and aggregative cells could be easily distinguished by plating on phloxine B-containing medium and observing colony and cellular morphologies (Fig. S1a). As shown in Fig. 6b–f and S10a–e, all tested evolved aggregative mutant strains outcompeted the WT yeast-form strain after 1 dpi in the brain, whereas only a subset of evolved aggregative mutant strains containing mutations in *BNI1*, *KIN3*, *APL2*, *NIP100*, and *ACE2* outcompeted the WT yeast-form strain in the kidney. Although yeast-form strain cells outcompeted most evolved aggregative mutant strain cells in the liver, lung, and spleen, the absolute colony-forming units (CFUs) in these organs were considerably lower than those in the brain and kidney. Overall, *C. auris* aggregative cells exhibited higher fitness levels relative to yeast-form cells during systemic infections. We note that although the host tissues were mechanically pulverized and homogenized for fungal burden and CFU assays, *C. auris* aggregates could not be entirely converted to the single-celled form. Thus, the fungal burdens we report for the aggregative mutant strains in our plating assays could be underestimated.

Consistent with our findings on the evolved aggregative mutant strains, our constructed aggregative mutant strains also exhibited higher fitness levels in the brain compared to the WT yeast-form control strain (Fig. 7). In addition, our constructed aggregative mutant strains displayed similar capacities to colonize the other host organs as the evolved aggregative mutant strains. Taken together, these results further indicate that aggregative cells of *C. auris* are better able to survive in the mouse brain during systemic infection compared to yeast-form cells.

## Aggregative cells are more resistant to phagocytosis by macrophages than yeast-form cells
Consistent with a previous study[36], we found that *C. auris* yeast-form cells can be efficiently phagocytosed by macrophages (using macrophage cell line RAW264.7), unlike aggregative cells (Fig. S11). Indeed, two representative evolved aggregative mutant strains (GZY127, *CHS1* p.Q826* mutant with pTDH3-*GFP* and GZY144, *BNI1* p.206 fs mutant with pTDH3-*GFP*) were resistant to phagocytosis compared to the WT yeast-form strain (GZY121, WT with pTDH3-*GFP*), which was efficiently engulfed by macrophages (Fig. S11). Typically, macrophages readily engulf fungal cells smaller than 10 μm in diameter. Thus, we hypothesized that the multicellular aggregates of *C. auris* are potentially too large to be engulfed by macrophages. To test this hypothesis, we converted the *C. auris* aggregates to single cells by sonication and tested them in our macrophage phagocytosis assay. Consistent with our hypothesis, these converted single cells, which were derived from *C. auris* aggregates, were efficiently engulfed by macrophages. This finding provides evidence to suggest that multicellular aggregates facilitate the ability of *C. auris* to escape phagocytosis by host innate immune cells due to their enlarged size.

## Aggregative cells are more resistant to host-derived antimicrobial peptides than yeast-form cells
Antimicrobial or host defense peptides are key components of the mammalian innate immune system[60]. We next tested whether the aggregative cells of *C. auris* have an advantage over single-celled yeast-form cells to survive in the host due to increased resistance to host-derived antimicrobial peptides. Human antimicrobial peptides LL-37 and PACAP display structural and antimicrobial similarities[61]. Both peptides have potent antifungal activity against pathogenic *Candida* species[61–63], and PACAP is selectively induced in the brain in response to bacterial or fungal infections[61]. We, therefore, examined the antifungal activity of LL-37 and PACAP against yeast-form cells and aggregative cells of *C. auris* in vitro. Propidium iodide (PI) staining and CFU assays were performed to evaluate the survival rates of the *C. auris* cells upon exposure to LL-37 and PACAP. After treatment with LL-37 and PACAP, nearly all (100%) off the WT yeast-form strain cells (BJCA001 and AR0386) were stained red with PI, whereas the ratio of PI-stained cells for the evolved aggregative mutant strain cells (FDAG4, *CHS1* p.Q826*; FDAG30, *BNI1* p.206 fs; FDAG9, *LRG1* p.K427*; FDAG1, *CAS4* p.1288 fs; and AR0386A, *SSD1* p.R549K) varied from 83% to 98% (Fig. 8a). We note that the survival rates of the different *C. auris* strains as determined by CFU assays largely correlated with the PI staining results (Fig. 8b, c). Taken together, these results demonstrate that *C. auris* aggregative cells are more resistant to the antifungal activity of host-derived antimicrobial peptides than yeast-form cells, which could contribute to their increased fitness during systemic infections.

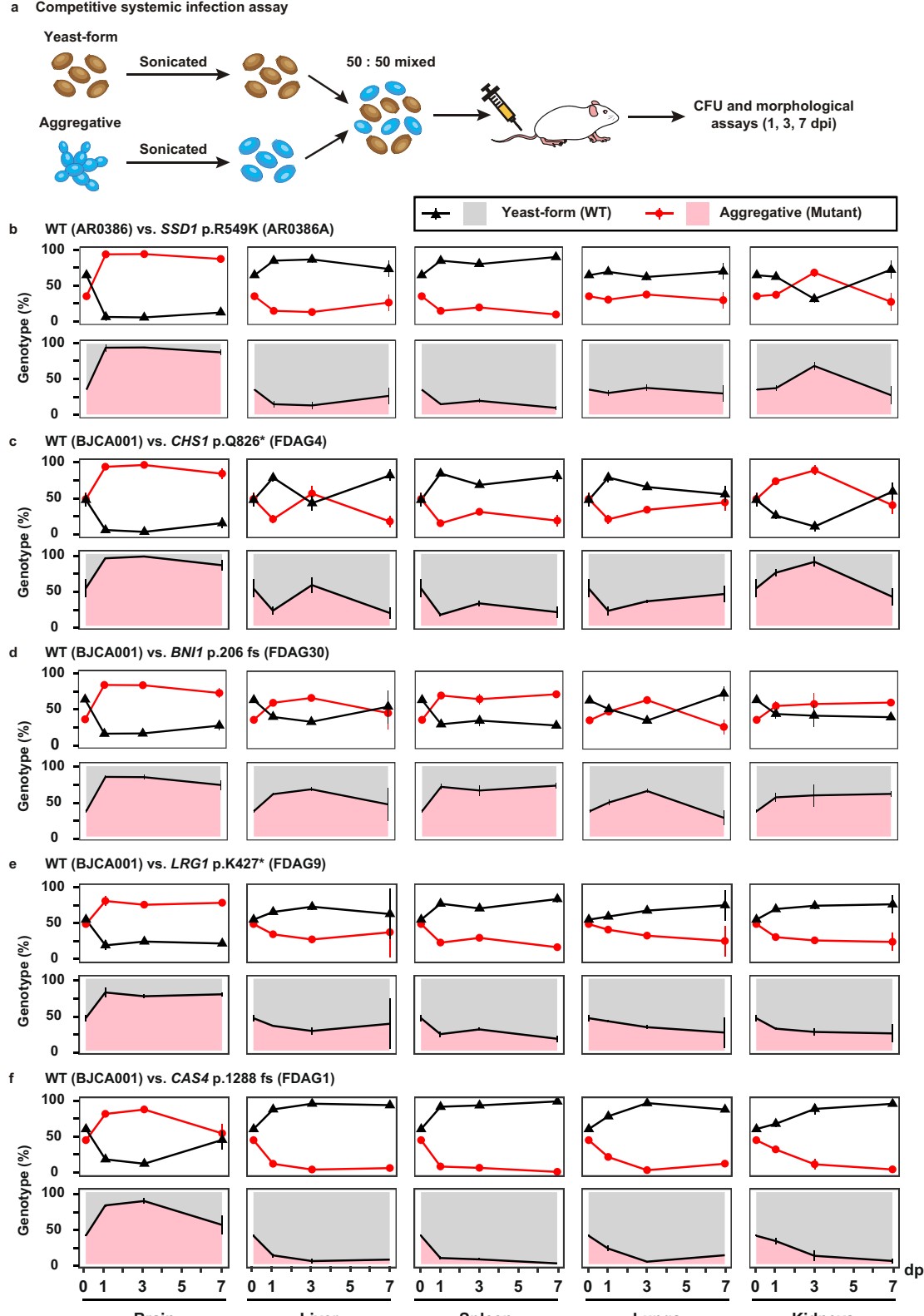

**a   Competitive systemic infection assay**

**b   WT (AR0386) vs. SSD1 p.R549K (AR0386A)**

**c   WT (BJCA001) vs. CHS1 p.Q826* (FDAG4)**

**d   WT (BJCA001) vs. BNI1 p.206 fs (FDAG30)**

**e   WT (BJCA001) vs. LRG1 p.K427* (FDAG9)**

**f   WT (BJCA001) vs. CAS4 p.1288 fs (FDAG1)**

Brain    Liver    Spleen    Lungs    Kidneys

## Host antimicrobial peptides LL-37 and PACAP promote the aggregative morphology through genetic mutations

Given that aggregative cells are more resistant to LL-37 and PACAP, we next wondered whether *C. auris* yeast-form cells could undergo rapid evolution into multicellular aggregative cells in the presence of these host-derived antimicrobial peptides. We, therefore, treated *C. auris* WT yeast-form cells with LL-37 or PACAP under in vitro culture conditions.

The fungal cells were then plated onto YPD plates containing the red dye phloxine B, several rough or red colonies grew on the plates ($10^{-5} \sim 10^{-4}$) and were isolated and subjected to microscopic assays. Thirty-three representative single colony isolates displaying the typical aggregative phenotype were then selected for whole genomic sequencing analysis. As shown in Fig. S12a, Table S4, and Dataset S1, similar to the host-induced aggregative mutant strains, the

**Fig. 6 | Competitive fitness and virulence assays of the *C. auris* yeast-form and aggregative cells. a** Schematic of the competitive infection assays. Yeast-form and aggregative cells were first subject to sonication. Equal numbers of single cells of the two morphologies ($5 \times 10^6$ yeast-form cells + $5 \times 10^6$ aggregative cells) were mixed and injected into the mice via the tail vein. At 1, 3, or 7 dpi, fungal cells were recovered from the brain, liver, spleen, lung, and kidney tissues, weighed, and pulverized using Zirconia beads (3.2 mm) with 60 Hz power for 120 s. The homogenized samples were then re-plated onto YPD medium plates containing the red dye phloxine B. CFU assays (per gram organ) were performed. Colonies formed by yeast-form and aggregative cells could be easily distinguished by their morphologies and coloration (**Fig. S1**). **b–f** Percentages of the yeast-form (WT) and aggregative (evolved mutant) strain cells in different mouse organs based on CFU assays (4 mice per strain were used for each time point). **b** WT (yeast-form, AR0386) versus *SSD1* p.R549K (AR0386A); **c–f** WT (yeast-form, BJCA001) versus *CHS1* p.Q826* (FDAG4), *BNI1* p.206 fs (FDAG30), *LRG1* p.K427* (FDAG9), and *CAS4* p.1288 fs (FDAG1). Data shown represents mean ± SD.

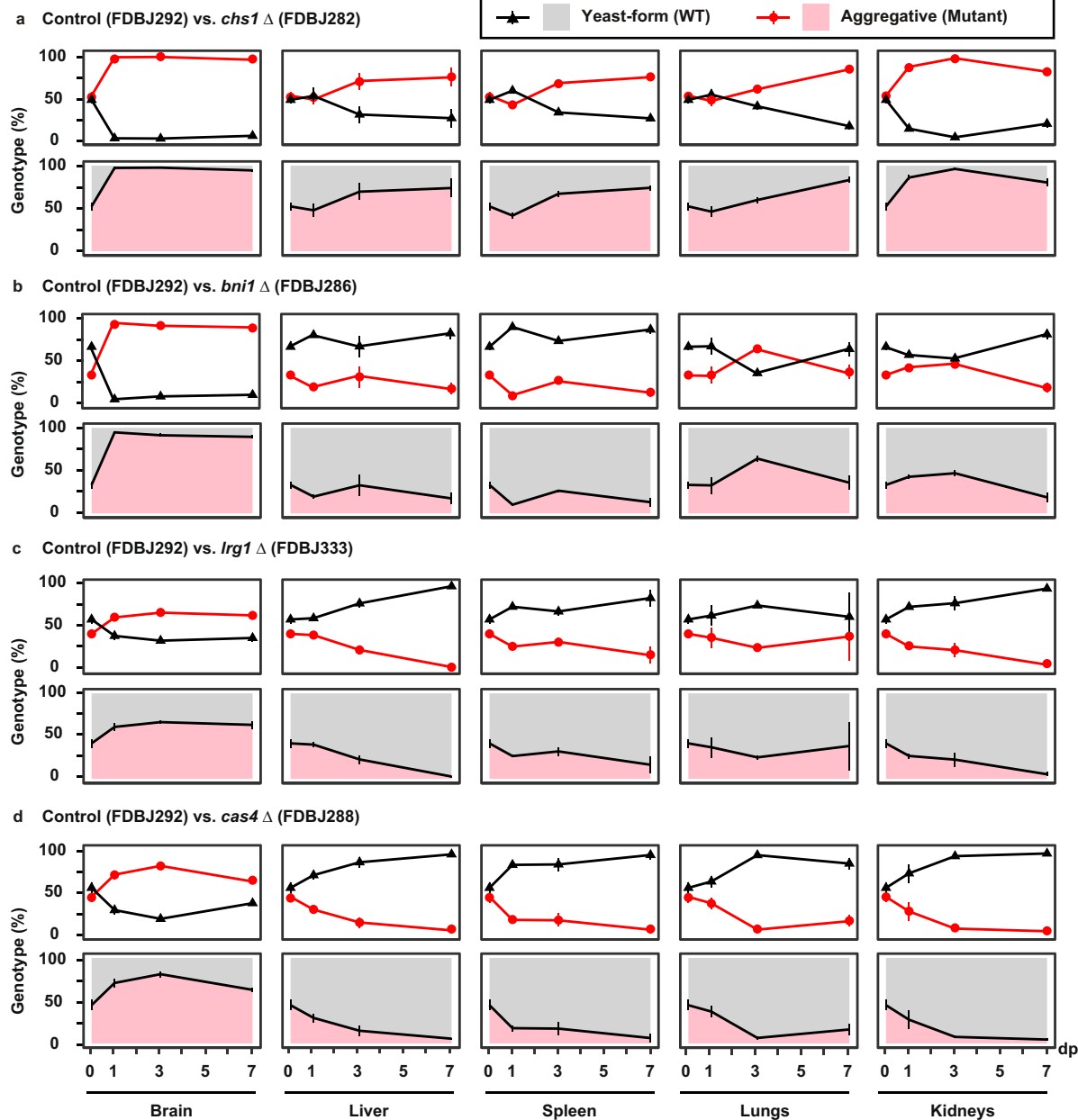

**Fig. 7 | Constructed mutant strains via deletion or amino acid substitution exhibited similar phenotypes and competitive fitness levels to those of the evolved strains. a–d** Percentages of *C. auris* cells of the control (yeast-form) and deletion mutant (aggregative) strains in different mouse organs based on CFU assays (4 mice per strain were used for each time point). Strains used: control (FDBJ292), *chs1-* (FDBJ282, **a**), *bni1-* (FDBJ286, **b**), *LRG1* p.K427* (FDBJ333, **c**), and *cas4* (FDBJ288, **d**). Competitive fitness and virulence assays were performed as described in Fig. 6. Line plot (upper graph) and area plot (lower graph) for each group are shown to display the proportions of yeast-form and aggregative cells. Two-tailed paired Student's *t*-tests were used to assess significance. Data shown represents mean ± SD.

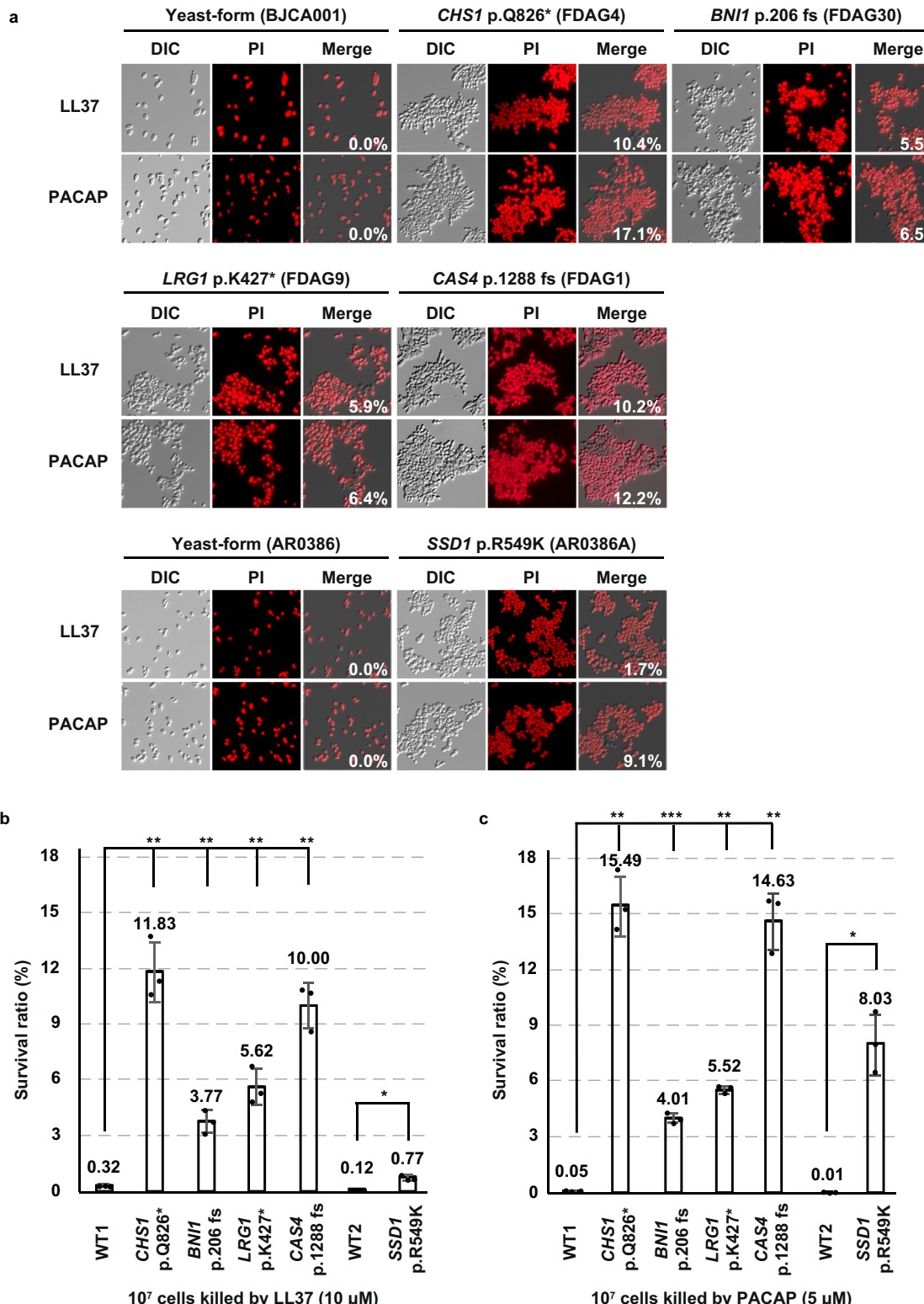

**Fig. 8 | Comparative analysis of in vitro susceptibility of *C. auris* yeast-form and aggregative cells to host antimicrobial peptides LL-37 and PACAP.** Strains used: BJCA001 (yeast-form), FDAG4 (*CHS1* p.Q826*), FDAG30 (*BNI1* p.206 fs), FDAG9 (*LRG1* p.K427*), FDAG1 (*CAS4* p.1288 fs), AR0386 (yeast-form), and AR0386A (*SSD1* p.R549K). **a** PI staining for antimicrobial killing assays. Yeast-form (BJCA001 and AR0386) or evolved aggregative (SSD1 p.R549K, CHS1 p.Q826*, BNI1 p.206 fs, LRG1 p.K427*, CAS4 p.1288 fs) strains (1 ×10⁷ cells/mL) were resuspended in 1 mM potassium phosphate buffer (PPB). *C. auris* cells were incubated with 10 μM LL-37 or 5 μM PACAP for 1 h at 37 °C. Treated cells were collected, washed, and stained with

PI. **b**, **c** Quantitative antimicrobial killing assays (n = 3). *C. auris* cells (1 ×10⁷ cells/mL) were resuspended in 1 mM PPB. Half of the cells were subjected to sonication and plated for CFUs. The other half of the cells were treated with 10 μM LL-37 (**b**) or 5 μM PACAP (**c**) for 1 h at 37 °C, respectively. Treated fungal cells were sonicated, diluted, and plated onto YPD medium for CFU analysis. The survival rate (%) of each strain was calculated. WT1: BJCA001, WT2: AR0386. The *P* value was determined by two-tailed paired Student's t-tests. *$P < 0.05$; **$P < 0.01$; ***$P < 0.001$. Data shown represents mean ± SD.

antimicrobial peptide-induced aggregative strains also carried mutations in genes involved in the regulation of cytokinesis, cellular polarity, and cell wall integrity. A number of mutated genes (CAS4, CBK1, LAA1, CHS1, HYM1, VRG4, ACE2, KIC1, LRG1, and VAN1) in the antimicrobial peptide-induced aggregative strains overlapped with those identified in the host-induced aggregative strains (Figs. 3 and S12). We note that in the absence of antimicrobial peptides, the spontaneous frequency of occurrence of the aggregative colonies was extremely low ($<1 \times 10^{-7}$). These results suggest that the aggregative morphology could have a fitness advantage under the stress of antimicrobial or host defense peptides that either induce genetic mutations in C. auris or suppress the growth of yeast-form cells.

## Discussion

Microevolution is a common adaptive strategy used by both prokaryotic and eukaryotic pathogens to generate genetic and phenotypic diversity to ultimately increase survival under changing environmental conditions[18–20,22,23,64–66]. Through adaptive microevolution under these changing conditions, cells within a population possessing beneficial mutations predominate the population in a short period of time. During host infection specifically, small-scale genetic changes and the resulting phenotypic plasticity often confer the pathogen with an increased ability to colonize or infect the host and to persist within the host under hostile conditions, such as while under attack by the host immune system. Experimental evolution studies in fungal pathogen cell populations have shown that exposure to or passage through the host leads to a higher rate of genetic mutations and phenotypic variations compared to cell populations cultured under in vitro conditions[19,65,66]. Uncovering the microevolutionary mechanisms of how fungal pathogens adapt to the host environment is central to our understanding of fungal biology and would provide important information to improve the treatment of fungal infections.

C. auris can exist in the typical yeast-form morphology but has also been reported to form unique multicellular aggregates in several recent studies[25,30–33]. The biological functions of these C. auris aggregates as well as the underlying regulatory mechanisms controlling their formation are poorly understood. In pathogenic fungi, morphological changes often provide invasive or survival advantages during host infection[16,67]. Given the prevalence of this aggregative morphology in clinical isolates of C. auris, it seems likely that this morphology should have advantages over the single-celled yeast-form morphology in the ever-changing host environment. Here, we report that passage through the mammalian host in a systemic infection model induces formation of multicellular aggregates in C. auris. We hypothesized that host-associated factors or environmental stresses could promote the formation of C. auris aggregative cells through adaptive microevolution. Indeed, the generation of small genetic variations such as point mutations and frameshift mutations were observed in a set of key genes required for cell division and/or budding. Compared to the single-celled yeast-form morphology, this aggregative morphology exhibits an overall increased fitness in the host and is exceptionally able to persist and survive in the brain. We note that in contrast to our findings in mice, Borman et al.[33] demonstrated that C. auris aggregative cells exhibited less pathogenicity compared to yeast-form cells in an invertebrate Galleria mellonella infection model[33]. This discrepancy between our results and those reported in Borman et al.[33] could be due to the different strains and infection models used. Overall, morphological plasticity not only affects virulence of C. auris but could also be a bet-hedging strategy to cope with environmental fluctuations during host infection. Given the biological and pathogenic differences of the yeast-form and aggregative cells of C. auris, our findings will have important clinical implications for the diagnosis and treatment of invasive C. auris infections.

Based on our findings, it is not clear why the aggregative cells of C. auris have a tissue tropism to the brain. There could be several possible reasons for this. First, aggregative cells, but not yeast-form cells, of C. auris could have a higher ability to cross the blood-brain barrier (BBB). Second, there are fewer immune cells (such as macrophages and neutrophils) in the brain than other tissues in the body. Third, there could be some specific and unidentified nutritional requirements necessary for the growth of aggregative cells of C. auris. This latter mechanism is reminiscent of the host inositol-mediated infections of the brain by C. neoformans[68]. Fourth, brain-specific inducers (e.g., the pituitary adenylate cyclase-activating polypeptide (PACAP) neuropeptide) could promote mutations in C. auris.

Although non-genetic- or epigenetic-induced morphological transitions in response to environmental changes or host-related factors are common in the major human fungal pathogens, such as C. albicans and C. tropicalis[11,16], phenotypic switches mediated by genetic mutations are rare in fungi. The host-induced genetic mutations in C. auris observed in this study could represent a new mechanism of host-driven microevolution and adaptation in emerging microbial pathogens. The host-induced mutated genes we observed in this study are involved in cell wall integrity, cytokinesis, cytoskeletal properties, and cellular polarization activities. Interestingly, their functions seem to converge on the process of cell division and/or budding, and thus result in the aggregative morphology in C. auris. Moreover, most mutated genes associated with the aggregative phenotype in C. auris are conserved in other yeast species such as C. albicans and S. cerevisiae, where they play similar roles in the regulation of filamentation and/or aggregation (Table S3).

We also found that despite minor differences among the C. auris evolved aggregative mutant strains, aggregative cells were in general less susceptible to macrophage phagocytosis and host antimicrobial peptides compared to yeast-form cells, indicating that this rapidly evolved aggregative morphology confers a survival advantage over the yeast-form morphology in the host setting. This strategy of morphological change via genetic alterations has also been observed in C. glabrata as well as other fungal pathogens in prior studies using experimental evolution experiments and pathogen-host interaction assays[18–20,22,23,65,66]. Moreover, some bacterial species can also switch to a multicellular filamentous morphology to survive in the host or other stressful environments[6], which confers bacterial cells with the ability to resist multiple unrelated insults such as host innate immune cell phagocytosis and exposure to antibiotics. Therefore, transitioning from a single-celled morphology to a multicellular morphology could be a general and multifactorial survival strategy for both bacterial and fungal pathogens.

A comparative analysis of the global transcriptional profiles of the yeast-form strain and four representative evolved aggregative isolates (CHS1 p.Q826*, BNI1 p.206 fs, LRG1 p.K427*, and CAS4 p.1288 fs mutants) demonstrates that many DEGs are functionally converged on the regulation of CWI, cell division, and cell polarity (Figs. 4 and S7). These results further reveal that the CWI, cytoskeleton, cytokinesis, and cell polarity pathways are inter-regulated and genetic perturbation of these pathways causes cell budding or division defects and formation of multicellular aggregates in C. auris.

In general, the C. auris aggregative morphology displayed higher fungal burdens and increased fitness over the yeast-form morphology during systemic infections. One particularly notable observation from our study is that aggregative cells of C. auris exhibited a strong tissue tropism to the brain, followed by the kidney in our systemic infection model (Figs. 5, 6, S8, S10, and S13). Consistent with this finding, aggregative cells caused much more severe brain tissue damage than yeast-form cells in our systemic infection model (Fig. S9). It remains to be determined how C. auris cells enter the brain and whether the observed mutations in the evolved aggregative strains occurred before or after the cells entered the brain.

We detected genetic mutations in all C. auris evolved aggregative strains, which are enriched in pathways involved in the regulation of

cell wall integrity, cytokinesis, cytoskeletal properties, and cellular polarization (Figs. 3 and S5). Since cell wall remodeling, cytokinesis, budding site selection, actin or actomyosin ring (AMR) contraction, and septum formation and destruction play critical roles in cell division or budding[48,55,69–71], genetic perturbation of these pathways could lead to cell division defects and the formation of multicellular aggregates. Several mutated genes present in the *C. auris* evolved aggregative strains identified in this study (e.g., *ACE2*, *BNI1*, and *CHS1*) have also been reported to be involved in the regulation of filamentous growth or aggregate formation in *C. auris* in other studies[36,37]. We also identified some of similar mutations associated with cell division and/or budding in many *C. auris* clinical isolates by analyzing existing *C. auris* genomic data available in the NCBI database (Fig. S5), suggesting that cellular aggregation caused by genetic mutations can be found in *C. auris* clinical isolates.

Another interesting finding in our study was that aggregative cells of *C. auris* are more resistant to phagocytosis by macrophages and host-derived antimicrobial peptides (Figs. 8 and S11). Since macrophages typically only engulf fungal cells of less than 10 μm in diameter[72], it is reasonable that the large size of the *C. auris* aggregates prevents them from being phagocytized. This phenomenon was also recently observed by several other groups[36,37]. We further found that the human-derived AMPs LL-37 and PACAP were able to induce genetic mutations in *C. auris*, leading to the formation of aggregates under in vitro culture conditions (Fig. S12). The mutations identified in the presence of these two AMPs were in genes encoding proteins enriched for involvement in cell division and/or cell budding processes (Figs. S4 and S10), implying that these AMPs could play roles in the generation of *C. auris* mutations in the mammalian host during systemic infections. Given that the expression of the endogenous AMP neuropeptide PACAP is selectively induced by bacterial and fungal pathogens in the brain, one possibility is that its expression could be associated with the accumulation of *C. auris* aggregative cells in the brain.

In summary, host-induced genetic and phenotypic diversity could facilitate the rapid evolution of *C. auris* during host infection. This mechanism could contribute to the recent emergence and rapid prevalence of *C. auris* worldwide. Our study provides an example of morphological transitions regulated by high frequency genetic mutations, which are usually controlled by non-genetic or epigenetic mechanisms in pathogenic fungi. Given the frequency of *C. auris* mutations observed during systemic infections with the host, the results of our study have important clinical implications in regard to the diagnosis and treatment of *C. auris* infections. Although relatively rare cases have been reported of *C. auris* infecting the brain to date, systemic/invasive *C. auris* infections could lead to the colonization of the brain. Uncovering the underlying molecular mechanisms of the development of this multicellular aggregative morphology in *C. auris* would improve our understanding of its survival strategy in the host and provide new insights into the development of antifungal therapeutics.

## Methods

### Ethics statement
All animal experiments were performed according to the guidelines approved by the Animal Care and Use Committee at Fudan University (2021JS004). The present study was approved by the Committee.

### *C. auris* strains and culture medium
Strains used in this study and detailed strain information are listed in Tables S1, S2, S4, and Dataset S1. YPD medium (2% Bacto peptone, 1% yeast extract, 2% dextrose, 2% agar, BD Becton Dickinson, Shanghai, China) was used for routine growth of *C. auris* strains. The red dye phloxine B (5 μg/mL, Sigma-Aldrich, Shanghai, China) was added to the medium for staining of the aggregative cells. *C. auris* cells were plated onto YPD medium plates and cultivated at 30 °C for 4 days. Colony and cellular morphologies were examined using a stereomicroscope (NSZ-810, Yongxin, China) and optical microscopes (DM2500 LED, Leica, Germany), respectively.

### Mouse systemic infection and fungal burden assays
Five- to six-week-old female BALB/c mice (weighing 16–18 g) were purchased from Vital River (Beijing, China) and used for all infection experiments. The mice were routinely maintained in a specific pathogen-free animal facility at a temperature of 21 °C, relative humidity of 50–70%, and under a constant 12-hour light/dark cycle. Mice were given free access to food and water throughout the study. Mice were allowed to acclimatize for 7 days before fungal inoculation.

For isolation of evolved *C. auris* aggregative strains, each mouse was injected with $4 \times 10^7$ cells of yeast-form cells through the tail vein. On day 3 after intravenous infection, the mice were killed and dissected. The brain, liver, spleen, lung, and kidney tissues were homogenized and plated onto YPD medium plates containing 5 μg/mL phloxine B (1000–1500 CFUs/plate). Wrinkled, pink, or red colonies were examined under a microscope. Aggregative isolates were subjected to whole genome sequencing (WGS).

For initial mutation screening from the brain, liver, spleen, lung, and kidney tissues, 28 mice were used for systemic infection (infected with 11 *C. auris* clinical isolates, Table S1). For comparative analyses in medium with and without phloxine B, brain tissue from 12 mice were used (infected with *C. auris* clinical isolate BJCA001, Table S2).

For single-strain virulence and fungal burden analysis, *C. auris* aggregative cells were first converted to single cells by sonication. Yeast-form or aggregative cells were sonicated for 30 s in 1 x PBS at 30% ultrasound amplitude using an ultrasonic homogenizer (195 W power, JY92-IIN, SCIENTZ, China). Approximately $1 \times 10^7$ *C. auris* cells in 200 μL $1 \times$ PBS were injected into each mouse through the tail vein. After 1, 3, or 7 dpi, the mice were weighed, sacrificed, and fungal burdens were assessed. For periodic acid-Schiff (PAS) staining and immunohistochemistry assays, mouse organs were fixed with formalin, embedded with paraffin, and sectioned. Anti-myeloperoxidase antibody (Abcam, Rabbit Source, ab208670, 1:1000 dilution) and anti-CD14 antibody (Abcam, Mouse Source, ab182032, 1:500 dilution) were used to demonstrate the recruitment of monocytes and neutrophils to the brain tissue, respectively.

For competitive fitness and virulence assays, *C. auris* yeast-form and aggregative cells were sonicated. A mixture of *C. auris* cells (50:50; $5 \times 10^6$ yeast-form + $5 \times 10^6$ aggregative cells in 200 μL $1 \times$ PBS) was injected into each mouse through the tail vein. After 1, 3, or 7 dpi, the mice were weighed, sacrificed, and fungal burdens were assessed. The brain, liver, spleen, lung, and kidney tissues were homogenized and plated onto YPD medium plates containing 5 μg/mL phloxine B.

To verify the yeast-form and aggregative phenotypes, the cellular morphology of at least 60 colonies for each strain was microscopically examined.

### Scanning electron microscopy (SEM) and transmission electron microscope (TEM) assays
SEM assays were performed as described in our previous publication[73]. Briefly, fungal cells were fixed with 2.5% glutaraldehyde at 4 °C overnight and washed with $1 \times$ PBS. Fungal cells were then dehydrated with ethanol (with gradually increasing concentrations: 50%, 75%, 90%, 100%, 100%, and 100%), dried, and coated with gold. Cell images were taken using a TM3000 SEM (Hitachi, Japan).

TEM assays were performed based on a previous report with slight modifications[74]. Fungal cells were frozen in liquid nitrogen using the EM ICE high pressure freezer (Leica, Germany). Frozen cells were then subjected to freeze substitution using the Leica EM AFS2 freeze substitution and low temperature embedding system for light and electron microscopy (Leica, Germany) in an environment with dried

acetone containing 2% osmium tetroxide and 0.1% uranyl acetate at −90 °C for 3 days. The samples were warmed to 4 °C steadily and then washed 3 times with acetone for 15 min each time. Next, the samples were infiltrated and embedded in Eponate 12 resin at 37 °C for 12 h and then at 65 °C for 48 h. The resin blocks were consecutively cut into 70 nm thickness using a Leica EM UC7 ultramicrotome (Leica, Germany), and the ultrathin sections were collected using 150 mesh formvar coated copper grids. The ultrathin sections were counterstained with 3% uranyl acetate for 10 min and then lead citrate for 5 min. The ultrathin sections were examined using a Talos L120C TEM (Thermo Fisher Scientific, America) at 120 KV.

### Whole genome sequencing and mutation analyses

Single colonies of *C. auris* strains were inoculated into liquid YPD medium and grown at 30 °C for 24 h. Fungal cells were then collected, and genomic DNA was extracted using the TIANamp Yeast DNA Kit (TianGen Biotech, Beijing, China) and short-insert fragment libraries were constructed according to the manufacturer's protocol. Whole genome sequencing (WGS) was performed using the PE150 sequencing and DNBseq tech platform (BGISEQ) and at least 3 GB of clean data per sample was generated. Raw genome sequence data is accessible from NCBI BioProject PRJNA1015296 and accession numbers are listed in Dataset S2. WGS assays were performed by BGI Genomics Co., Shenzhen, China. Genomic data of 4482 *C. auris* strains available in the NCBI SRA database (https://www.ncbi.nlm.nih.gov/sra, Dataset S2) was downloaded and used for analysis. Whole genome analysis was performed according to our previous publication[25]. Briefly, the raw reads were trimmed to remove low-quality (phred score ≤ 10), ambiguous and adaptor bases using the FASTX-Toolkit v0.0.14 (http://hannonlab.cshl.edu/fastx_toolkit/index.html). The clean reads were mapped to the genomic assembly of *C. auris* strain B8441 (NCBI accession number: GCA_002759435.2) using BWA mem 0.7.17 software with default settings[75]. SAMTools v1.361[76], Picard Tools v1.56 (http://picard.sourceforge.net), and GATK v2.7.2 were used for single nucleotide variant (SNV) and InDel analyses[77]. The sorted BAM datasets were used for copy number variation (CNV) analysis of all coding genes with command "samtools depth" across certain gene regions. An amino acid mutations were annotated using ANNOVAR[78]. Gene ontology (GO) analysis was performed by R package GOplot[79].

### Macrophage phagocytosis and cytotoxicity assay

RAW264.7 cells (ATCC, TIB-71) were cultivated in Dulbecco's Modified Eagle's Medium (DMEM, Gibco, Shanghai, China) supplemented with 10% FBS and 1% penicillin-streptomycin in 5% $CO_2$ at 37 °C. Macrophage phagocytosis assays were performed as described previously[74]. RAW264.7 cells ($1 \times 10^5$) were grown in glass bottom cell culture dish (Φ 20 mm) for 24 h. RAW264.7 cells were then washed with 1 × PBS. *C. auris* cells ($1 \times 10^5$, with a GFP reporter under the control of the *TDH3* promoter) were resuspended in 1 mL DMEM and mixed and co-incubated with RAW264.7 cells (MOI = 1:1) for 60 min.

### Host-associated antimicrobial peptide treatment assays

For killing assays, the WT (BJCA001 and AR0386) and five mutant strains FDAG4 (CHS1 p.Q826*), FDAG30 (BNI1 p.206 fs), FDAG9 (LRG1 p.K427*), FDAG1 (CAS4 p.1288 fs), and AR0386A (SSD1 p.R549K) were used. We selected these strains because the corresponding mutated genes are involved in the major signaling pathways or biological processes identified in our study (Fig. 3). *C. auris* yeast-form and aggregative cells were grown to logarithmic growth phase. $1 \times 10^7$ CFU/mL in 1 mM potassium phosphate buffer (PPB) were treated with 10 μM LL-37 (APExBIO, Shanghai, China) or 5 μM PACAP 1–38 (ACMEC, Shanghai, China) human antimicrobial peptides for 1 h at 37 °C. Treated cells were then collected, washed, and stained with propidium iodide (PI). To analyze the survival rate, half of the cell culture was subjected to sonication and CFU assays. Fungal cells were sonicated for 30 s in 1 x

PBS at 30% ultrasound amplitude using an ultrasonic homogenizer (JY92-IIN, SCIENTZ, China) and the sonicated suspension were then diluted and plated on YPD medium for CFU analysis. The other half was treated with 10 μM LL-37 or 5 μM PACAP 1–38 for 1 h at 37 °C and then subject to sonication and CFU analysis. Survival rate (%) = (CFU/mL after AMP treatment) / (CFU/mL before AMP treatment) x 100%.

To isolate *C. auris* aggregative mutant strains, yeast-form cells ($1 \times 10^6$ cells/mL) were treated with 5 μM LL-37 or 1 μM PACAP 1–38 in 1 mM PPB for 1 h at 37 °C. *C. auris* cells were then diluted and plated on YPD medium containing 5 μg/mL phloxine B. After 4 days of growth at 30 °C, wrinkled, pink, or red colonies containing aggregative cells were subjected to microscopy and whole genome sequencing (WGS) analysis.

### Generation of *C. auris* mutant strains

To facilitate deletion of the genes identified from the evolved aggregative mutant strains, we first constructed a *C. auris his1Δ* mutant using BJCA001 as the parental strain[80] and pSFS2A/*caSAT1* as the deletion plasmid[81]. The upstream and downstream flanking sequences of *C. auris HIS1* were amplified from genomic DNA of strain BJCA001 with primer pairs FDBJpr1 (with *Apa*I site)/FDBJpr2 (with *Xho*I site) and FDBJpr3 (with *Sac*II site)/FDBJpr4 (with *Sac*I site). The PCR products were then cloned into plasmid pSFS2A, which contains a *caSAT1* positive selection marker[81]. Linearized plasmid was then transformed into strain BJCA001 to replace the *HIS1* gene, generating the *his1Δ* mutant strain. Correct transformants were first examined by verifying the loss of the target region by PCR with primers FDBJpr5 and FDBJpr6. Another two PCR reactions were then performed to verify correct genomic integration using primer pairs FDBJpr9/FDBJpr10 and FDBJpr7/FDBJpr8 targeting the knockout cassette (selective marker) and the 3′- or 5′-flanking regions of the *HIS1* gene (Fig. S4a). The *caSAT1* flipper cassette was then excised by growth on YPM medium (1% yeast extract, 2% peptone, and 2% maltose) as previously reported[81], generating the *his1Δ* mutant strain (FSR1319).

The fusion PCR product recombination strategy[82] was used to delete *ACE2, CAS4, CHS1, BNI1, HOF1, IQG1, APL2, APM1, LRG1, SSD1, VAN1, OCH1, NIP100*, and *SGS1* genes in the *C. auris his1Δ* mutant (FSR1319). Primer pairs (Maker-FWD/Maker-REV) were used to amplify the *HIS1* marker expression cassettes from plasmid pSN52-CauHIS1 by PCR. To delete these genes of interest, strain FSR1319 was transformed with the fusion PCR products of the *CauHIS1* flanked by 5′- and 3′-flanking fragments of the target genes and cultured on SCD amino acid dropout plates for prototrophic selection growth. As described for *HIS1* deletion, correct transformants were first examined by verifying the loss of the target region by PCR. And another two PCR reactions were then performed to verify the correct genomic integration using checking primers (Fig. S4b). The PCR products were sequenced. The cellular and colony morphologies were then examined to confirm their similar phenotypes. Two to four independent transformants for each gene mutation were used for morphological or virulence assays. A similar fusion PCR product recombination strategy was used to construct the *HIS1*-reconstituted strain in *C. auris* strain FSR1319. Primer pairs (Maker-FWD/Maker-REV) were used to amplify the *HIS1* marker expression cassettes from plasmid pSN52-CauHIS1. And the 5′- and 3′-flanking fragments of *C. auris HIS1* were amplified from genomic DNA of strain BJCA001. All primers used for PCR are listed in Dataset S4.

### Plasmid construction

The plasmid pTDH3[30] was used for the construction of plasmid pTDH3-GFP. To ectopically express GFP in *C. auris* strains BJCA001, FDAG4 (*CHS1* p.Q826*) and FDAG30 (*BNI1* p.206 fs), the coding region of GFP was amplified from plasmid pNIM1[83] by PCR using primers Cau-GFP-EcoRV-F and Cau-GFP-XhoI-R. PCR products were digested with restriction enzymes *Eco*RV and *Xho*I, and then subcloned into the *Eco*RV/*Xho*I site of plasmid of pTDH3, generating plasmid pTDH3-GFP.

The plasmid pTDH3-GFP was linearized with enzyme *Stu*I, and the linearized plasmid pTDH3-GFP was transformed into strains BJCA001, FDAG4, and FDAG30.

To construct the plasmid pSN52-*CauHIS1*, primer pairs FSP240 (with a *Bam*HI site)/FSP241 and FSP242/FSP243 (with a *Not*I site) were used to amplify the TEF promoter and TEF terminator from the template plasmid pSN52, respectively. The *C. auris HIS1* gene was amplified from genomic DNA of strain CBS12766 using primer pair FSP926/FSP927. The fusion PCR product of TEF promoter-HIS1-TEF terminator was then subcloned into plasmid pSN52 at the *Bam*HI/*Not*I site[82], generating plasmid pSN52-*CauHIS1*. All primers used for plasmid construction are listed in Dataset S4.

### *C. auris* transformations
The transformation method was performed according to a previous protocol of *C. albicans* transformation with slight modifications[84]. Briefly, *C. auris* cells were first grown to the mid-logarithmic phase (-OD1.8) in 25 mL of YPD medium with constant shaking. The culture was centrifuged in a 50 mL Falcon tube at 1500 x g at 4 °C for 5 min and washed with precooled sterile water. Fungal cells were resuspended in 4 mL TE-LiAc buffer (10 mM Tris, 1 mM ethylenediaminetetraacetic acid, 100 mM lithium acetate) and incubated at 30 °C for 1.5 hs. DTT (200 μL 1 M) was then added to the cell suspension and the cells were incubated at 30 °C for another 0.5 h with shaking. Fungal cells were washed with precooled sterile water 3 times, then washed with precooled 1 M sorbitol one time, and then resuspended in 120 μL 1 M sorbitol. *C. auris* competent cells (40 μL) were mixed with 1–5 mg of the deletion cassette for transformation by electroporation (MicroPulser™, Bio-Rad, Shanghai, China). After electroporation, fungal cells were resuspended in YPD and incubated at 30 °C with constant shaking for 2 h and then plated onto selectable medium plates for 3–4 days of growth at 30 °C.

### RNA-Seq assays
*C. auris* cells were initially grown on YPD plates for 5 days. About 500 fungal cells were then spotted onto YPD plates and incubated at 30 °C for 3 days (the cultures were in exponential phase). Three biological repeats were performed for each strain. Fungal cells were harvested and washed with ddH$_2$O and total RNA was extracted for RNA-Seq analysis. The quantity of RNA was assessed using measuring the OD at 260 nm and 280 nm by a Nanodrop-2000 Ultraviolet Spectrophotometer (Thermo Fisher, China). Each RNA sample had an A260:A280 ratio between 1.8 and 2.0. RNA integrity was evaluated using the Agilent 2200 Tape Station (Agilent Technologies, USA, RNA integrity number above 7.0). The libraries were sequenced using the Illumina NovaSeq platform according to the company's protocol (performed by Berry Genomics Co., Beijing, China). Approximately 3 GB of data were obtained by sequencing each library. The raw RNA-Seq data is available in the NCBI BioProject PRJNA1015296 and SRA accession numbers are listed in DataSet S2. Low-quality (Phred score 10), ambiguous, and adaptor bases were removed from the raw reads using the FASTX-Toolkit v0.0.14 (http://hannonlab.cshl.edu/fastx_toolkit/index.html). The clean reads were aligned to the reference genome of *C. auris* (NCBI accession number GCA_002759435.2) with the software HiSat2 v2.0.5 with default parameters. Transcriptional expression of different samples was estimated with StringTie v1.3.3b using default parameters[85]. Differentially expressed genes were analyzed by R package DESeq2[86]. The Euclidean distance was used to assess the association between the RNA-Seq data. Principal Component Analysis (PCA) was analyzed using the script plotPCA from DESeq2. Differentially expressed genes must satisfy two criteria: (i) a fold change value higher than or equal to 1.5; (ii) an adjusted p-value (false discovery rate [FDR]) lower than 0.05. All data were deposited in the Sequence Read. Predicted protein interaction analysis were performed with STRING web tools (https://cn.string-db.org/cgi) and the network was visualized with R package networkD3.

### Statistics and reproducibility
Two-tailed paired Student's *t*-tests and log rank tests were used for statistical analyses as stated in the figure legends. *C. auris* colony and cellular morphologies, brain lesion, and immunohistochemistry images presented in the figures were representative data of at least three independent experiments.

### Reporting summary
Further information on research design is available in the Nature Portfolio Reporting Summary linked to this article.

## Data availability
The authors declare that the data supporting the findings of this study are available within the article and its Supplementary Information files. Genomic sequence and RNA-seq data have been deposited in the SRA database and the accessible numbers are listed in Dataset S2. Prediction of protein interaction assays were based on STRING database (https://cn.string-db.org/). Source data are provided in this paper. Source data are provided with this paper.

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

## Acknowledgements

This work was supported by the National Key Research and Development Program of China (grant no. 2021YFC2300400 to G.H. and no. 2022YFC2303000 to J.B.), National Natural Science Foundation of China (award 31930005 and 82272359 to G.H. and nos. 32170193 and 32000018 to J.B.), the National Institutes of Health (NIH) National Institute of General Medical Sciences (NIGMS) (grant R35GM124594 to C.J.N.), and by the Kamangar family in the form of an endowed chair (to C.J.N.). The content is the sole responsibility of the authors and does not represent the views of the funders. The funders had no role in the design of the study; in the collection, analyses, or interpretation of data; in the writing of the manuscript; or in the decision to publish the results.

## Author contributions

J.B., Z.G., T.Z., C.J.N., C.C., H.C., and G.H. conceived and designed the study; J.B., Z.G., T.Z., C.L.E., H.C., and G.H. performed the data analysis and wrote the manuscript; J.B., Z.G., T.Z. conducted all of the experiments; J.B., Z.G., T.Z., C.L.E., and G.H. contributed methodology, formal analysis; J.B., Z.G., T.Z., C.L.E., C.J.N., C.C., H.C., and G.H. discussed the experiments and results; J.B., C.J.N., and G.H. contributed validation, funding acquisition.

## Competing interests

Clarissa J. Nobile is a cofounder of BioSynesis, Inc., a company developing diagnostics and therapeutics for biofilm infections. All other authors declare no conflicts of interest.
