## [Peer Review File · Nature Communications]

Rapid evolution of an adaptive multicellular morphology of *Candida auris* during systemic infectionREVIEWER COMMENTS

Reviewer #1 (Remarks to the Author):

The manuscript entitled "Rapid evolution of an adaptive multicellular morphology of *Candida auris* during systemic infection in the host" describes the identification and characterization of a wide array of mutations that arise in *C. auris* during in vivo infection and that result in an altered growth state involving multicellular clusters that appear to increase virulence. Generally, the work described appears significant and of interest to the audience of *Nature Communications*, and the language of the manuscript is clear. However, there are multiple major concerns that compromise the interpretation of the results, particularly with concerns for details pertaining to isolates and strains used for specific experiments, and assumptions that appear to confound some experimental design as currently described. Because of the significant number of concerns that compromise the interpretation of the results or make reproducing the experiments as described difficult/ not possible, my recommendation is to reject this manuscript for publication at this time. Concerns/ questions are listed below:

Major concerns:

Throughout the manuscript- The isolates and strains used in each experiment are not clearly defined. 12 isolates are listed in the Table S1, but which isolates or related strains used in each experiment and the rationale for this choice needs to be clearly stated. For example in lines 188 to 193 the description of the isolation of multicellular strains from in vivo experiments does not include which *C. auris* isolates were used and the rationale for their selection. This seems like an essential piece of information and should not be overlooked.

Lines 290 to 295- the mutant strains constructed are insufficiently described. How many strains were generated for each mutation? Were there independent biological replicates? How were the strains confirmed? There is no mention of any form of verification in the methods. This is very concerning. Further, why was phloxine B used in selecting transformants? The dominant marker drug should be enough to select for the manipulation of interest, adding selection with phloxine B adds a bias to find a phenotype rather than the genotype of interest and is fundamentally a flawed experimental design. This combined with the lack of description of replicates or strain verification significantly call into question the experiments using these strains as well.

The assumption that phloxine B only stains multicellular growth forms of *C. auris* ("the red dye phloxine B, which exclusively stains colonies containing aggregative cells pink or red" lines 190 to 192) seems to underpin several experiments, including the selection of in vivo evolved strains (lines 189 to 193) and the differentiation of cells in the competition assay (lines 412 to 414). This is a faulty assumption as *C. auris* cells growing in a unicellular form can appear red/ pink on media supplemented with 5ug/mL phloxine B. This has been reported in publications, including some by authors on this manuscript, where pink growth on phloxine B was reported to be associated with MLT α mating type and unicellular growing pink colonies from the isolate BJCA001 (which appears to have been a key isolate in this work) were documented. As pink growth on phloxine B alone is thus not a clear differentiating phenotype of colonies, selecting based primarily this phenotype could introduce a bias. If an approach was taken to avoid this bias, it should be more clearly described.

The authors state that "Our results indicate that a single passage within the mammalian host is sufficient to induce the formation of this multicellular aggregative morphology." This appears to be an underlying observation to the entire manuscript, but it is insufficiently supported. The authors state 96 rough or pink/red colonies were obtained. How many of the 12 clinical isolates in Table S1 were used in this experiment? How many colonies were screened to find the 96 strains? Both wrinkled and phloxine staining colonies occur naturally at a low frequency under standard laboratory culturing conditions. How does the rate of occurrence/ emergence of these colonies differ in vivo from under standard in vitro culturing conditions. No data is clearly presented. Thus the authors' statement seems insufficiently supported by the evidence provided.

The design of the genetic manipulations as shown in Figure S3B seem to overlook/ fail to

acknowledge the potential confounding factor that design not only introduces the mutation of interest, but also leaves a FRT site and flanking sequence around the multiple cloning sites from the pSFS2 between the 5' and 3' homology regions as well. Based on the description provided it is unclear if this "scar" of the manipulation is in the ORF as well or more likely the promoter of the gene of interest. This design feature should be clarified and acknowledged as a possible confounder/ complication of the design and the sequence alteration should be confirmed by sequencing the final strains. If this was done it should be described.

Several experiments are insufficiently described for the experiments to be repeated/ results verified by others. For example, in the results section "Aggregative cells are more resistant to host-derived antimicrobial peptides than yeast-form cells" the comparison is being made between the wildtype yeast form cells (undisclosed isolate background) and multicellular strains (unknown mutants from in vitro evolution experiments and/ or genetic manipulations). This should be clearly described. What mutants were assessed in this experiment and what was the rationale for their selection.

Accession numbers for WGS and RNAseq experiments must be made publicly available. The raw data should be uploaded to a public repository like SRA and the accession numbers provided.

Moderate concerns:

Results section "C. auris clinical isolates exhibit yeast-form and aggregative morphologies" starting at line 170- This section of the results seems more like background. Information described summarizes findings from five or so prior publications and the only data presented is a set of images of one of 12 clinical isolates used in this study paired with what appears to be a single derivative strain in the same background. This background information can be combined with the following section or perhaps expanded to include a comparison/ evaluation/ description of additional isolate backgrounds.

The 40 distinct mutations described in lines 213 to 216 seem insufficiently described. How did isolate background influence the mutations observed? Were any of the mutations associated with a single background assessed. Did all 96 strains screened have one of these 40 mutations?

Transformation methods are insufficiently described. What method of transformation was used? (Electroporation/ chemical) This is a key aspect of the methodology that should have been included in the methods.

The RNAseq experiments are described as each sample yielding 3 billion reads (lines 771 to 774). This is approximately 2 orders of magnitude higher than standard sequencing depth for comparable RNAseq experiments. Is this a typo or was there a reason to sequence so incredibly deep?

Why was RNA seq performed on cells harvested from YPD spot plates after 3 days of growth? After 72 hours it would be likely the cells harvested are in or near stationary growth phase and transcriptionally different than cells more actively growing. Is there a rationale for this timing decision?

Reviewer #2 (Remarks to the Author):

This report reveals important information about the aggregation phenotype of *Candida auris*, and suggests that this is the result of the accumulation of mutations that occur the course of infection that induce cellular aggregation in tissue of internal organs – in particular the brain. They show the evolution of mutated phenotypes that compromise cell separation and lead to changes in the cell wall that resist the action of antifungal peptides and macrophage mediated phagocytosis. They show that the strains isolated from the brain have a number of mutations that are linked to a cell

wall, morphogenesis and polarity gene network, that could contribute the aggregation phenotype. They provide evidence that the aggregating strain has a competitive advantage over the parental non-aggregating parent strain in being able to colonise the brain (and to a lesser extent other organs). They recapitulate the brain-tropic phenotype in isogenic mutants that they make which mimic the phenotype of the spontaneous mutant evolved strains.

This is a very interesting and comprehensive study that significantly advances the field. In a number of respects the data are surprising and the manuscript could be improved by addressing a number of key questions that emerge from their study.

One issue is that this is very much an experimental study in mice. The links and comparisons clinical infection in humans are not as well integrated as they might be. For example, *C. auris* is really adapted for skin colonisation and it would seem likely that evolution takes place on the skin. In the relatively rare cases of invasive disease, infection in brain is unusual and more commonly infection of the liver and kidneys is more common. To my knowledge brain infections have been infrequently reported in cases when the patient has been in a neurosurgery ward where contamination into the brain may be a higher risk. So in humans there seems to be no evidence for a brain tropism as reported here in mice.

The correlation between the ability to aggregate and higher virulence is taken as being expected. But one study using a wax moth model suggested that the non-aggregating strains are the more pathogenic (Borman AM, Szekely A, Johnson EM. Comparative Pathogenicity of United Kingdom Isolates of the Emerging Pathogen *Candida auris* and Other Key Pathogenic *Candida* Species. *mSphere*. 2016 Jul-Aug;1(4).). This should be mentioned.

Note that most of the study seems to have been done using one aggregating *C. auris* strain. Aggregating strains are found in multiple clades that are phylogenetically divergent. A table is shown with strains across the *C. albicans* clades but what evidence is there that the study is generalizable across the various *C. auris* clades?

Of the 27 mutations that are identified the most frequent is in the chitin synthase gene *CHS1*. It is important to say what class of *CHS* chitin synthase this is. In *C. albicans* *CHS1* is a class II enzyme that makes the primary chitinous septum. It is essential, and the mutant had to be created by regulated conditional repression of a remaining wild type allele in a heterozygous background (Munro et al., 2001). The *C. albicans chs1* mutant is the only one that forms chains of cells – although the morphology is distinct – generating progressively enlarged budding cells that eventually spontaneously lyse. In contrast they report that “the p.Q826* mutation had a relatively minor impact on cell physiology” (line 344-345). They should therefore say what class *CHS1* is (perhaps it is a class I enzyme that is not essential in most fungi) they should measure the chitin content of the mutant and show TEMs of the septa to see if the primary septum is missing or aberrant in a way that could explain the lack of cytokinesis.

The suggestion that aggregation is due to a failure to undergo normal cell division could be placed more clearly in the context of published literature. For example, Santana and O’Meara (2021) showed that septation may be affected in aggregating strains of *C. auris* through mutation of the *Ace2* transcription factor that regulates chitinase (*Cht1*) that is required for septal plate dissolution. (The equivalent study in *Cryptococcus* is also mentioned). Malavia et al (2023) suggest on the basis of microscopy, that septation in aggregating strains of *C. auris* is complete and normal, and that aggregation occurs post-cell division but that there is an enrichment of amyloid forming protein genes in aggregating strains. Pelletier et al (2023) suggest that “two different types of aggregation: one induced by antifungal treatment which is a result of a cell separation defect; and a second which is controlled by growth conditions and only occurs in strains with the ability to aggregate”. [<https://doi.org/10.1101/2023.04.21.537817>doi: bioRxiv preprint]. These observations are not assimilated in the manuscript.

Given that they find that: (a) aggregates are difficult to phagocytose by macrophages (also shown elsewhere) and; (b) aggregates accumulate in tissues. This suggests that single (non-aggregated) yeast cells disseminate in the bloodstream and that aggregation takes place in the tissues. Did they ever look for yeast cells in the blood to confirm this? In lines 589-591 they say this has not

been done, but given the nature of the experiments I am sure they must have had samples from blood from the mice to look at. When they create the equivalent mutants they would presumably be constitutively aggravating and they will therefore aggregate in the bloodstream. Did this happen? I might have expected that this might compromise efficient dissemination in the bloodstream.

It is also surprising that many of the mutations lie in pathways that regulate important or even essential processes (morphogenesis, cell wall formation, polarity, RAM pathway etc) and therefore would be expected to cause a fitness defect or low burdens in tissues. Some of the equivalent mutations in other yeasts are lethal or attenuated in virulence (e.g. *chs1*). This is interesting because here they show that most of the spontaneous mutants in this study have a positive competitive fitness compared to WT - and also enhanced brain tropism relative to wild type. (Although this tropism was not seen for all tissues). Is this not surprising given that nature of the mutations?

Most of these mutations have been characterised in a number of yeast species – for example *Candida albicans* and *Saccharomyces cerevisiae*. They should take time to compare the phenotypes (including but going beyond the aggregation phenotype) in their evolved mutated strains and those described for other yeast species in publications in the literature. A table of comparisons with equivalent mutations in other yeast pathogens would be helpful if making such comparisons.

Also without obvious explanation is the observation that exposure to antimicrobial peptides induced that same subset of mutations. They suggest that the presence of the peptides actually “induce” mutations (lines 499-501). “These results provide evidence that antimicrobial or host defense peptides could induce genetic mutations in *C. auris* and play potential roles in the rapid evolution of the observed aggregative morphology.”

Against this seems very surprising. Surely the most likely scenario would be a Darwinian mechanism of selection of spontaneous mutations that have a fitness advantage. Suggesting that the peptides are mutagenic and induce genetic mutations directly without direct evidence difficult to support. If they were genuinely mutagenic would you not expect the entire genome to be mutagenized across all gene sets? The authors say that “as expected a number of mutated genes...in the antimicrobial-peptide induced aggregative strains overlapped with those identified in the host-induced aggregative strains” (lines 493-496). Why would this be expected? Has the induction of mutations by such peptides been observed previously? If not why not?

Minor points:

The Introduction has quite a long section on phenotypic adaptation in *Candida albicans* (lines 88-115) and other fungi and organism (to line 137). This seems unnecessarily detailed and is rather a digression from the focus of the paper.

A number of key publications have not been cited or are not fully integrated.

For example, a highly relevant paper describes a neutropenic mouse model, where large aggregates of *C. auris* cells were found in hearts, kidneys and liver of all mice suggesting that most strains of *C. auris* form aggregates in tissues. (Forgacs, L., Borman, A.M., Prepost, E., Toth, Z., Kardos, G., Kovacs, R., Szekely, A., Nagy, F., Kovacs, I., Majoros, L., 2020. Comparison of in vivo pathogenicity of four *Candida auris* clades in a neutropenic bloodstream infection murine model. *Emerg. Microb. Infect.* 9, 1160–2119.)

Reviewer #3 (Remarks to the Author):

Bing et al submitted their manuscript dealing with morphological and genetical evolution of *Candida auris* cells in systemic mouse model. The manuscript contains several valuable data regarding *C. auris* adaptation and evolution both at physiological and molecular levels. As main conclusion, the manuscript provides an example of morphological transitions regulated by different

mutations; furthermore, it revealed several underlying molecular mechanisms of the evolution of aggregative morphology. However, there are more shortages which should be addressed:

-Line 188-192 The way of presentation is not clear. Are they pooled data or one mouse derived data. It would be worth to examine the number of pink colonies in case of each organs per mouse and present the obtained values as mean and SD in case of different organs. Moreover, we did not receive any information about the number of used mice.

-Line 195 What is the reason of tissue tropism? Hypothesis? Please write about it in discussion.

-Line 275 In Figure 3, please write in the legend that the presented genes are independent genes.

-Line 225-256 What do the numbers in parenthesis mean? Clarify them!

-Line 234 *Saccharomyces cerevisiae* instead of *S. cerevisiae*. It is the first mention.

-Line 252 „Seven genes“ Please list them!

-Line 255 „number of genes“ How much? Exact number please!

-Line 274 „9 types of mutations“ Please list them!

-Line 374 „cell by sonication prior to tail vein injection“ Did you perform quantitative control culturing in order to exclude the potential sonication related harmful effect?

-Line 482-488 This part belongs to material and methods, please remove from here.

-Line 636 Manufacturer of YPD medium? Please include.

-Line 638 Manufacturer of phloxine B? Please include.

-Line 645 „six-week-old mice“? Why did you use six-week-old mice? Instead of ages please write weight! It is more indicative. I think, based on ages, animals are too young. Were they males or females? Please include the permission number of animal experiments.

-Line 649 Please write the exact mouse number used in experiments!

-Line 658 „ultrasound amplitude“ Please write the exact wavelength!

-Line 705 Write the detailed whole genome analysis!

-Line 715 Manufacturer of DMEM? Please include.

-Line 717-724 „Approximately“ is too vague. Please modify the problematic sentences.

-Line 729 What was the protocol of sonication and CFU assays? Please clarify!

-Line 765-786 The description of RNA-seq analysis is not too detailed. The background of analysis and the analysed data needs to be better explained. Some suggestions: principal component analysis (PCA) should be included. How did you check the quality of RNA, what were the criteria? How did you perform downstream analysis? Please write more details about the evaluation of transcriptome data, definition of gene categories etc.

-Discussion: Generally, I suggest that Authors rewrite and/or reedit the discussion because it many times repeats the previously described facts and observations in Results without further detailed explanations/hypothesis/extrapolation etc. Moreover, it has crucial importance that the observed results should be translated into clinical aspect regarding *C. auris* infection.

Response to reviewers' comments

Our point-to-point response to reviewers' comments is highlighted in blue.

Reviewer #1 (Remarks to the Author):

The manuscript entitled "Rapid evolution of an adaptive multicellular morphology of *Candida auris* during systemic infection in the host" describes the identification and characterization of a wide array of mutations that arise in *C. auris* during in vivo infection and that result in an altered growth state involving multicellular clusters that appear to increase virulence. Generally, the work described appears significant and of interest to the audience of Nature Communications, and the language of the manuscript is clear. However, there are multiple major concerns that compromise the interpretation of the results, particularly with concerns for details pertaining to isolates and strains used for specific experiments, and assumptions that appear to confound some experimental design as currently described. Because of the significant number of concerns that compromise the interpretation of the results or make reproducing the experiments as described difficult/ not possible, my recommendation is to reject this manuscript for publication at this time.

The authors thank this reviewer for the positive summary and for pointing out weaknesses in the experimental design of our initial submission. In the revised revision, we have addressed these issues.

Concerns/ questions are listed below:

Major concerns:

Throughout the manuscript- The isolates and strains used in each experiment are not clearly defined. 12 isolates are listed in the Table S1, but which isolates or related strains used in each experiment and the rationale for this choice needs to be clearly stated. For example in lines 188 to 193 the description of the isolation of multicellular strains from in vivo experiments does not include which *C. auris* isolates were used and the rationale for their selection. This seems like an essential piece of information and should not be overlooked.

In the revised revision, we updated the strain list tables (**Tables S1, S2, S4, and Dataset S1**). We now provided detailed information for each evolved aggregative mutants (including parental strains, references or sources for the parental strains, numbers, names, and genotype of evolved aggregative mutants, and citations in the figures). Also, we added the detailed strain information to the figures and figure legends.

The rationale for strains used for in vivo experimental evolution assays was to cover the four major genetic clades and to examine the general feature of host-induced mutations in *C. auris* (**Table S1**). **Page 10, lines 210-212**

Lines 290 to 295- the mutant strains constructed are insufficiently described. How many strains were generated for each mutation? Were there independent biological replicates? How were the strains confirmed? There is no mention of any for of verification in the methods. This is very concerning. Further, why was phloxine B used in selecting transformants? The dominant

marker drug should be enough to select for the manipulation of interest, adding selection with phloxine B adds a bias to find a phenotype rather than the genotype of interest and is fundamentally a flawed experimental design. This combined with the lack of description of replicates or strain verification significantly call into question the experiments using these strains as well.

We thank this reviewer for pointing this out. We revised the methods section (**Pages 39-40, lines 837-875**). Two to four independent transformants (strains) for each mutation were generated. Cellular and colony morphologies were examined to confirm that their phenotypes were similar. The mutants were initially identified by examining the loss of the target region by PCR. We then followed the standard in the field and performed another two PCR reactions to verify their correct integration into the genome using a primer targeting the knockout cassette (selective marker) and another primer targeting the 3'- or 5'-flanking regions of the deleted gene (**Figure S3**). The PCR products were then sequenced.

To avoid potential bias, we redesigned and performed gene knockout assays in the revised manuscript. In these assays, we did not add the red dye phloxine B to the medium to obtain the new mutants. Briefly, we used a more efficient and more acceptable method in the *Candida* field to delete *C. auris* genes. A *his1*Δ reference strain was generated in clinical strain BJCA001. The *CauHIS1* disruption marker cassette and a fusion PCR protocol were then used to generate knockout mutations in the *his1*Δ strain (this method was adapted from the *Candida albicans* knockout strategy, *Noble and Johnson AD. Eukaryot Cell. 2005*).

Page 39-40, Figure S4

The assumption that phloxine B only stains multicellular growth forms of *C. auris* ("the red dye phloxine B, which exclusively stains colonies containing aggregative cells pink or red" lines 190 to 192) seems to underpin several experiments, including the selection of in vivo evolved strains (lines 189 to 193) and the differentiation of cells in the competition assay (lines 412 to 414). This is a faulty assumption as *C. auris* cells growing in a unicellular form can appear red/ pink on media supplemented with 5ug/mL phloxine B. This has been reported in publications, including some by authors on this manuscript, where pink growth on phloxine B was reported to be associated with MLTa mating type and unicellular growing pink colonies from the isolate BJCA001 (which appears to have been a key isolate in this work) were documented. As pink growth on phloxine B alone is thus not a clear differentiating phenotype of colonies, selecting based primarily this phenotype could introduce a bias. If an approach was taken to avoid this bias, it should be more clearly described.

We corrected the statement about phloxine B to be more accurate. In the revised manuscript, we stated that **"this dye was used in the screening assay because it largely stains colonies containing aggregative cells of *C. auris* pink or red and facilitated our abilities to identify evolved mutant strains."** **Page 10, lines 214-217**

To see whether the addition of phloxine B would cause a bias for the selection of in vivo evolved strains, we performed comparative experiments in the revised manuscript. The brain tissue of *C. auris* infected mice was ground and plated onto two types of YPD agar plates (one with and the other without phloxine B, please see the below figure). **Page 11, lines 232-240**

Figure: Comparative analysis of isolation of evolved mutant strains on YPD medium with and without phloxine B. Some mutant strains could not be identified on the medium without phloxine B.

We found that many more evolved aggregative *C. auris* mutant strains could be identified from the plates with phloxine B dye than from plates without the dye. In the presence of phloxine B, mutant strains exhibiting strong colony phenotypes as well as relatively weak colony phenotypes could be identified. However, only those mutant strains showing strong colony phenotypes (such as rough and wrinkled ones) could be identified in the absence of phloxine B. Therefore, in the selection assay of in vivo evolved strains, we believe that the addition of phloxine B is beneficial in facilitating the efficient identification of evolved strains and should not cause a bias. If anything, the selection of evolved aggregative strains in the absence of phloxine B could cause a potential bias towards identifying mutant strains with strong colony phenotypes.

As shown in the upper figure, to verify the phenotype of the mutant strains, we examine their cellular morphologies once we identify the potential mutant strains with colony phenotypes.

For the competition assays (Figures 6, 7, S10), the addition of phloxine B would greatly increase the efficiency and accuracy of the differentiation assay of *C. auris* yeast-form and aggregative cells. Again, to verify the mutant strains, we examined the cellular morphologies of at least 60 colonies for each experiment. Moreover, we found that it was technically difficult to discriminate between the reference strain and certain evolved mutant strains in the competition assays in the absence of phloxine B. **Pages 10 and 35**

In summary, we believe that the red phloxine B dye is best used for screening and competition assays but not for constructing gene deletion strains.

The authors state that "Our results indicate that a single passage within the mammalian host is sufficient to induce the formation of this multicellular aggregative morphology." This appears to

be an underlying observation to the entire manuscript, but it insufficiently supported. The authors state 96 rough or pink/red colonies were obtained. How many of the 12 clinical isolates in Table S1 were used in this experiment? How many colonies were screened to find the 96 strains? both wrinkled and phloxine staining colonies occur naturally at a low frequency under standard laboratory culturing conditions. How does the rate of occurrence/ emergence of these colonies differ in vivo from under standard in vitro culturing conditions. No data is clearly presented. Thus the authors statement seems insufficiently supported by the evidence provided.

We clarified this point in the revised manuscript and re-described the conclusion sentence ("Our results indicate that ..."). **Pages 11 lines 230-231**

In total, we obtained 113 colonies with morphology changes. From them, 96 strains from the five organs were initially identified (**Table S1**) and 17 were identified from the brain tissue in our revised experiments (comparative analysis with and without phloxine B, **Table S2**). For the initial 96 evolved isolates, all 11 clinical isolates in **Table S1** were used for animal experiments. Four mice were used for infection with each strain BJCA001, CBS10913, and AR0386, and two mice were used for infection with the other 8 isolates. Of note, one clinical strain in **Table S1** of our initial submission was not used in animal experiments.

For comparative assays with and without phloxine B, 12 mice were used for infection with strain BJCA001. The same amount of brain tissue from each mouse was used for plating on the medium with and without phloxine B. **Page 33 lines 729-733**

The highest and average frequency of aggregative mutant strains from the brain were 0.057% and 0.011%, respectively, whereas the average frequency from the liver, spleen, lung, and kidney tissues were 0.0019%-0.0086%. No evolved mutant strains were isolated from the blood of infected mice. In total, we analyzed about 1,800,000 colonies (1,000-1,500 colonies/plate; over 1,600 plates were analyzed). **Page 10 lines 217-221**

The mutation rate (frequency of wrinkled and phloxine stained colonies) under the standard in vitro culturing condition (YPD medium) was < 0.0001% (not observed in our experiments). Of note, the mutation frequency varied in the different mice and different parent *C. auris* strains.

The design of the genetic manipulations as shown in Figure S3B seem to overlook/ fail to acknowledge the potential confounding factor that design not only introduces the mutation of interest, but also leaves a FRT site and flanking sequence around the multiple cloning sites from the pSFS2 between the 5' and 3' homology regions as well. Based on the description provided it is unclear if this "scar" of the manipulation is in the ORF as well or more likely the promoter of the gene of interest. This design feature should be clarified and acknowledged as a possible confounder/ complication of the design and the sequence alteration should be confirmed by sequencing the final strains. If this was done it should be described.

The pSFS2A/caSAT1 (developed by the Joachim Morschhäuser lab) and fusion PCR with amino acid nutritional selectable marker (developed by the Johnson lab and Noble lab) strategies are the two most widely used strategies for gene deletion in *Candida* species. Both methods have advantages and disadvantages.

We agree with this reviewer that the FRT site and flanking sequence from the pSFS2A plasmid could affect cellular physiology. Also, as mentioned by this reviewer in an earlier point,

there was a phloxine B issue in our initial submission. To address this, we redesigned our gene knockout assays and used a *his1Δ* reference strain and fusion PCR strategy in the revised manuscript (adapted from the strategy reported by *Noble and Johnson AD, Eukaryot Cell. 2005*). We found that this *CauHIS1*/fusion PCR knockout strategy was much more efficient than the pSFS2A/caSAT1 method in the deletion of *C. auris* genes. For example, it was difficult to delete the whole ORF region of *LRG1*, *APL2*, and *SSD1* using the pSFS2A/caSAT1 method. However, we efficiently deleted the whole ORF regions of these genes using the *HIS1*/fusion PCR strategy in the revised manuscript. More importantly, the morphological and virulence features of the new mutant strains were similar to those obtained by the pSFS2A/caSAT1 deletion assay, suggesting that both deletion assays are effective. **Figure S4 Pages 39-40**

Several experiments are insufficiently described for the experiments to be repeated/ results verified by others. For example, in the results section "Aggregative cells are more resistant to host-derived antimicrobial peptides than yeast-form cells" the comparison is being made between the wild type yeast form cells (undisclosed isolate background) and multicellular strains (unknown mutants from in vitro evolution experiments and/ or genetic manipulations). This should be clearly described. What mutants were assessed in this experiment and what was the rationale for their selection.

We clarified these issues in the revised manuscript. **Page 24, 25; lines 519-523, 545-547, Figures 8 and S11 legend.**

We selected these strains (FDAG4, *CHS1* p.Q826*; FDAG30, *BNI1* p.206fs; FDAG9, *LRG1* p.K427*; FDAG1, *CAS4* p.1288fs; and AR0386A, *SSD1* p.R549K) for the experiments because the mutations represent the major signaling pathways or biological processes identified in our study (**Figure 3**).

Accession numbers for WGS and RNAseq experiments must be made publicly available. The raw data should be uploaded to a public repository like SRA and the accession numbers provided.

The accession numbers for WGS and RNAseq experiments are listed in **Dataset S2** and this information was also mentioned in the method section (**Pages 36, 42**). We have now also deposited the data at the NCBI SRA database.

Moderate concerns:

Results section "C. auris clinical isolates exhibit yeast-form and aggregative morphologies" starting at line 170- This section of the results seems more like background. Information described summarizes findings from five or so prior publications and the only data presented is a set of images of one of 12 clinical isolates used in this study paired with what appears to be a single derivative strain in the same background. This background information can be combined with the following section or perhaps expanded to include a comparison/ evaluation/ description of additional isolate backgrounds.

As suggested, we combined this paragraph with the following section.

The 40 distinct mutations described in lines 213 to 216 seem insufficiently described. How did isolate background influence the mutations observed? Were any of the mutations associated

with a single background assessed. Did all 96 strains screened have on of these 40 mutations?

In the revised manuscript, we provided detailed information of each evolved mutant in the supplementary **Tables S1, S2, S4**, and **Dataset S1**. Also, we described the detailed information of each strain in the corresponding figures and figure legends.

Transformation methods are insufficiently described. What method of transformation was used? (Electroporation/ chemical) This is a key aspect of the methodology that should have been included in the methods.

We used the electroporation method for *C. auris* transformation. The method was added to the revised manuscript.

(Pages 41-42 lines 895-911)

The RNAseq experiments are described as each sample yielding 3 billion reads (lines 771 to 774). This is approximately 2 orders of magnitude higher than standard sequencing depth for comparable RNAseq experiments. Is this a typo or was there a reason to sequence so incredibly deep?

This was indeed a typo. We corrected this mistake in the revised methods section. **(Page 43, lines 925)**

Why was RNA seq performed on cells harvested from YPD spot plates after 3 days of growth? After 72 hours it would be likely the cells harvested are in or near stationary growth phase and transcriptionally different than cells more actively growing. Is there a rationale for this timing decision?

C. auris cells grow slower than *C. albicans* cells. The growth rate also depends on the strain background, culture medium, temperature, and initial inoculum size. Based on our growth assays, we found that *C. auris* cells were still in mid-exponential phase after 72 hours of incubation. As shown below, we were unable to obtain enough cells from the 24-hour or 48-hour cultures for sequencing. **Page 42, lines 914-916**

Figure: Growth of *C. auris* strains on YPD medium. Strains used: BJCA001 (yeast-form), CHS1p.Q826*, BNI1p.206fs, LRG1p.K427*, and CAS4p.1288fs. Approximately 500 cells of each strain were spotted onto YPD medium and cultured at 30°C for five days. (a) Colony morphology. Scale bar = 4mm. (b) Time course CFU assay.

Reviewer #2 (Remarks to the Author):

This report reveals important information about the aggregation phenotype of *Candida auris*, and suggests that this is the result of the accumulation of mutations that occur the course of infection that induce cellular aggregation in tissue of internal organs – in particular the brain. They show the evolution of mutated phenotypes that compromise cell separation and lead to changes in the cell wall that resist the action of antifungal peptides and macrophage mediated phagocytosis. They show that the strains isolated from the brain have a number of mutations that are linked to a cell wall, morphogenesis and polarity gene network, that could contribute the aggregation phenotype. They provide evidence that the aggregating strain has a competitive advantage over the parental non-aggregating parent strain in being able to colonise the brain (and to a lesser extent other organs). The recapitulate the brain-topic phenotype in isogenic mutants that they make which mimic the phenotype of the spontaneous mutant evolved strains.

This is a very interesting and comprehensive study that significantly advances the field. In a number of respects the data are surprising and the manuscript could be improved by addressing a number of key questions that emerge from their study.

The authors thank this reviewer for the nice summary and helpful suggestions.

One issue is that this is very much an experimental study in mice. The links and comparisons clinical infection in humans are not as well integrated as the might be. For example, *C. auris* is really adapted for skin colonisation and it would seem likely that evolution takes place on the skin. In the relatively rare cases of invasive disease, infection in brain is unusual and more commonly infection of the liver and kidneys is more common. To my knowledge brain infections have been infrequently reported in cases when the patient has been in a neurosurgery ward where contamination into the brain may be a higher risk. So in humans there seems to be no evidence for a brain tropism as reported here in mice.

We now discuss this point in the revised manuscript. (Pages 32, lines 694-699)

We believe that this study has implications for clinical practices. First, many clinical *C. auris* isolates carried the mutations identified in this study and many previous studies reported aggregative strains isolated from clinical settings. Second, we identified a large set of strains containing the mutations identified in this study based on our analysis of publicly available genomic sequences. Although relatively rare cases of *C. auris* brain infection have been reported to date, systemic/invasive infections could lead to the colonization of the brain. A possible reason for the rare cases of brain infection reported could be due to difficulty in diagnosing *C. auris* infections (especially for the *C. auris* aggregative strains).

We performed skin colonization experiments but failed to isolate the aggregative mutants. The host stresses (such as antimicrobial peptides LL-37 and PACAP and oxidative stress) could be required for the induction of this aggregative phenotype. In unpublished data, we found that a major component of skin care products can induce mutations in *C. auris*. This is ongoing work that we plan to publish as a separate paper since it is outside of the scope of this study.

The correlation between the ability to aggregate and higher virulence is taken as being expected. But one study using a wax moth model suggested that the non-aggregating strains are the more pathogenic (Borman AM, Szekely A, Johnson EM. Comparative Pathogenicity of United Kingdom Isolates of the Emerging Pathogen *Candida auris* and Other Key Pathogenic *Candida* Species. *mSphere*. 2016 Jul-Aug;1(4).). This should be mentioned.

Thanks for providing this reference.

We cite this article in the revised manuscript and discussed this point. Page 27, lines 589-594

Note that most of the study seems to have been done using one aggregating *C. auris* strain. Aggregating strains are found in multiple clades that are phylogenetically divergent. A table is shown with strains across the *C. auris* clades but what evidence is there that the study is generalizable across the various *C. auris* clades?

To clarify this point, we provided three Tables (S1, S2, and S4) and Dataset S1 containing detailed strain information in the revised manuscript. The references/sources for the parental strains, numbers, names, and genotypes of evolved aggregative mutants, and citations in the figures are presented.

We performed in vivo infection experiments using 11 isolates of the four major genetic clades (of note, one clinical strain in Table S1 of our initial submission was not used for animal

experiments). Although there could be a difference in mutation frequency, we obtained evolved aggregative mutants from all 11 of the parental strains (revised **Table S1** and **Table S2**). These findings indicate that the host-induced rapid evolution is a general phenomenon across different *C. auris* clades.

We also added a supplementary figure to the revised manuscript (**Figure S3**), which demonstrates evolved mutants of the same genes that derived from different parental strains (different genetic clades) exhibited a similar phenotype.

Page 14, lines 307-310

Of the 27 mutations that are identified the most frequent is in the chitin synthase gene *CHS1*. It is important to say what class of CHS chitin synthase this is. In *C. albicans* *CHS1* is a class II enzyme that makes the primary chitinous septum. It is essential, and the mutant had to be created by regulated conditional repression of a remaining wild type allele in a heterozygous background (Munro et al., 2001). The *C. albicans chs1* mutant is the only one that forms chains of cells – although the morphology is distinct - generating progressively enlarged budding cells that eventually spontaneously lyse. In contrast they report that “ the p.Q826* mutation had a relatively minor impact on cell physiology” (line 344-345). They should therefore say what class *CHS1* is (perhaps it is a class I enzyme that is not essential in most fungi) they should measure the chitin content of the mutant and show TEMs of the septa to see if the primary septum is missing or aberrant in a way that could explain the lack of cytokinesis.

Thanks for pointing this out. Similar to its *C. albicans* counterpart, *C. auris CHS1* encodes a class II enzyme. We mention and discuss this point in the revised manuscript. We also successfully deleted this gene again using the *HIS1*/fusion PCR assays. It is still unclear why this gene is not essential in *C. auris*. We also tried to delete the *CHS1* gene in *C. albicans* but failed to obtain mutants.

We did not establish the chitin content test assay yet but performed TEM assays to see the septa (as shown below and in **Figure 2**). We indeed observed aberrant septa in *C. auris* aggregative cells. However, it is still hard to say whether the primary septum is missing or aberrant based on our data. **Page 12, lines 263-264**

Figure: TEM assays were performed using strain *CHS1* p.Q826* (FDAG4).

The suggestion that aggregation is due to a failure to undergo normal cell division could be placed more clearly in the context of published literature. For example, Santana and O'Meara (2021) showed that septation may be affected in aggregating strains of *C. auris* through mutation of the Ace2 transcription factor that regulates chitinase (Cht1) that is required for septal plate dissolution. (The equivalent study in *Cryptococcus* is also mentioned). Malavia et al (2023) suggest on the basis of microscopy, that septation in aggregating strains of *C. auris* is complete and normal, and that aggregation occurs post-cell division but that there is an enrichment of amyloid forming protein genes in aggregating strains. Pelletier et al (2023) suggest that “two different types of aggregation: one induced by antifungal treatment which is a result of a cell separation defect; and a second which is controlled by growth conditions and only occurs in strains with the ability to aggregate”. [<https://doi.org/10.1101/2023.04.21.537817>doi: bioRxiv preprint]. These observations are not assimilated in the manuscript.

As suggested, we cite and discuss these references in the revised manuscript.

Pelletier et al. (2023), Malavia-Jones et al. (2023) and our (2023, PLoS Pathogens) studies indicate that there are two types of aggregative morphologies. One type is formed due to a defect in cell division, while the other is associated with the overexpression of adhesion-associated genes.

Pages 8 and 13

Given that they find that: (a) aggregates are difficult to phagocytose by macrophages (also shown elsewhere) and; (b) aggregates accumulate in tissues. This suggest that single (non-aggregated) yeast cells disseminate in the bloodstream and that aggregation takes place in the tissues. Did they ever look for yeast cells in the blood to confirm this? In lines 589-591 they say this has not been done, but given the nature of the experiments I am sure they must have had samples from blood from the mice to look at. When they create the equivalent mutants they would presumably be constitutively aggravating and they will therefore aggregate in the bloodstream. Did this happen? I might have expected that this might compromise efficient dissemination in the bloodstream.

Yes, we examined the fungal burden in the blood. Compared to the brain ($5 \times 10^5 - 1.6 \times 10^6$ CFUs/g) and other tissues ($4 \times 10^4 - 3 \times 10^6$ CFUs/g), the fungal burden in the blood was extremely low (250 -1,000 CFUs/mL). We did not identify evolved aggregative mutants from the blood. The blood environment could be harsh for *C. auris* cells and most fungal cells would be rapidly killed in the bloodstream.

Page 10 lines 217-218

It is also surprising that many of the mutations lie in pathways that regulate important or even essential processes (morphogenesis, cell wall formation, polarity, RAM pathway etc) and therefore would be expected to cause a fitness defect or low burdens in tissues. Some of the equivalent mutations in other yeasts are lethal or attenuated in virulence (e.g. *chs1*). This is interesting because here they show that most of the spontaneous mutants in this study have a positive competitive fitness compared to WT - and also enhanced brain tropism relative to wild type. (Although this tropisms was not seen for all tissues). Is this not surprising given that nature of the mutations?

Yes, this result is surprising. However, these mutations could be common in clinical isolates based on our analysis of publicly available genomic sequences. Also, we recently identified an *ACE2* mutation in two clinical isolates from China (Tian S, Bing, J., et al., *Front Microbiol.* 2023 Jun 7;14:1174878). These results suggest that *C. auris* could adopt a strategy of mutating to adapt to environmental changes (unlike *C. albicans*, in which non-genetic changes regulate morphological changes).

Page 13 lines 277-281

In addition, we added a table to the revised manuscript that shows the conserved function of these genes or pathways in yeast species *C. auris*, *C. albicans*, and *S. cerevisiae* (Table S3). Given the conserved feature of these genes in the regulation of cell aggregation, filamentous or invasive growth in these yeast species, it is reasonable that they play a similar role in the regulation of morphological changes in *C. auris*.

Page 116 lines 335-338

Most of these mutations have been characterised in a number of yeast species – for example *Candida albicans* and *Saccharomyces cerevisiae*. They should take time to compare the phenotypes (including but going beyond the aggregation phenotype) in their evolved mutated strains and those described for other yeast species in publications in the literature. A table of comparisons with equivalent mutations in other yeast pathogens would be helpful if making such comparisons.

As suggested, we added a table to the revised manuscript. We compared the function of these mutated genes and discussed this point (Table S3). **Page 16 lines 335-338**

Also without obvious explanation is the observation that exposure to antimicrobial peptides induced that same subset of mutations. They suggest that the presence of the peptides actually “induce” mutations (lines 499-501). “These results provide evidence that antimicrobial or host defense peptides could induce genetic mutations in *C. auris* and play potential roles in the rapid evolution of the observed aggregative morphology.” Against this seems very surprising. Surely the most likely scenario would be a Darwinian mechanism of selection of spontaneous mutations that have a fitness advantage. Suggesting that the peptides are mutagenic and induce genetic mutations directly without direct evidence difficult to support. If they were genuinely mutagenic would you not expect the entire genome to be mutagenized across all gene sets? The authors say that “as expected a number of mutated genes...in the antimicrobial-peptide induced aggregative strains overlapped with those identified in the host-induced aggregative strains” (lines 493-496). Why would this be expected? Has the induction of mutations by such peptides been observed previously? If not why not?

This is interesting, and we agree with this reviewer on this point.

We reworded this statement for clarity. We did not expect that antimicrobial peptides could induce the same mutations. There could be many other mutation inducers in the host.

We also revised the sentence “These results provide evidence that antimicrobial or host...”.

“These results suggest that the aggregative morphology could have an advantage under the stress of antimicrobial or host defense peptides that either induce genetic mutations in *C. auris* or suppress the growth of yeast-form cells.” **Pages 25-26, lines 544-552**

Minor points:

The Introduction has quite a long section on phenotypic adaptation in *Candida albicans* (lines 88-115) and other fungi and organism (to line 137). This seems unnecessarily detailed and is rather a digression from the focus of the paper. A number of key publications have not been cited or are not fully integrated. For example, a highly relevant paper describes a neutropenic mouse model, where large aggregates of *C. auris* cells were found in hearts, kidneys and liver of all mice suggesting that most strains of *C. auris* form aggregates in tissues. (Forgacs, L., Borman, A.M., Prepost, E., Toth, Z., Kardos, G., Kovacs, R., Szekely, A., Nagy, F., Kovacs, I., Majoros, L., 2020. Comparison of in vivo pathogenicity of four *Candida auris* clades in a neutropenic bloodstream infection murine model. *Emerg. Microb. Infect.* 9, 1160–2119.)

Thanks for this suggestion. We cite and discuss these papers in the revised Introduction section. **Page 8 lines 169-172**

Reviewer #3 (Remarks to the Author):

Bing et al submitted their manuscript dealing with morphological and genetical evolution of *Candida auris* cells in systemic mouse model. The manuscript contains several valuable data regarding *C. auris* adaptation and evolution both at physiological and molecular levels. As main conclusion, the manuscript provides an example of morphological transitions regulated by different mutations; furthermore, it revealed several underlying molecular mechanisms of the evolution of aggregative morphology.

The authors thank this reviewer for this nice summary.

However, there are more shortages which should be addressed:

–Line 188-192 The way of presentation is not clear. Are they pooled data or one mouse derived data. It would be worth to examine the number of pink colonies in case of each organs per mouse and present the obtained values as mean and SD in case of different organs. Moreover, we did not receive any information about the number of used mice.

In total, we obtained 113 evolved aggregative strains (96 initially obtained + 17 during manuscript revision). 28 mice were used for initial identification of aggregative mutants and 12 for the later assay (from the brain tissue). Since the average values of identified mutants were extremely low (0 -10 isolates per mouse), we do not think that it is necessary to provide the mean and SD values in this case.

Pages 34, lines 729-733, Figure 2 legend

–Line 195 What is the reason of tissue tropism? Hypothesis? Please write about it in discussion. Thanks. We discuss this point in the revised Discussion section. **Page 28 lines 600-610**

–Line 275 In Figure 3, please write in the legend that the presented genes are independent genes.

Done. These genes were independently identified from different evolved aggregative mutants.

–Line 225-256 What do the numbers in parenthesis mean? Clarify them!

Clarified. **Page 12-13, lines 266-268**

–Line 234 *Saccharomyces cerevisiae* instead of *S. cerevisiae*. It is the first mention.

Corrected.

–Line 252 „Seven genes” Please list them!

As suggested, we list them in the revised manuscript: *Pan1, Arc19 and AP adaptors (Apl2, Apl4, Laa1, Apm1, and Bsp1*.

–Line 255 „number of genes” How much? Exact number please!

Provided.

–Line 274 „9 types of mutations” Please list them!

Revised.

–Line 374 „cell by sonication prior to tail vein injection” Did you perform quantitative control culturing in order to exclude the potential sonication related harmful effect?

Yes. The control yeast-form cells were subject to the same sonication treatment. We also examined the CFUs before and after sonication.

–Line 482-488 This part belongs to material and methods, please remove from here.

Revised.

–Line 636 Manufacturer of YPD medium? Please include.

Included.

–Line 638 Manufacturer of phloxine B? Please include.

Included.

–Line 645 „six-week-old mice”? Why did you use six-week-old mice? Instead of ages please write weight! It is more indicative. I think, based on ages, animals are too young. Were they males or females? Please include the permission number of animal experiments.

We provide detailed information on the mice used in the revised manuscript.

Five- to six-week-old female BALB/c mice (weighting 16-18 g) were used.

–Line 649 Please write the exact mouse number used in experiments!

Provided.

–Line 658 „ultrasound amplitude” Please write the exact wavelength!

Provided.

–Line 705 Write the detailed whole genome analysis!

Revised accordingly.

Pages 36-37

–Line 715 Manufacturer of DMEM? Please include.

Provided.

–Line 717-724 „Approximately” is too vague. Please modify the problematic sentences.

Revised.

–Line 729 What was the protocol of sonication and CFU assays? Please clarify!

Provided.

Page 38

–Line 765-786 The description of RNA-seq analysis is not too detailed. The background of analysis and the analysed data needs to be better explained. Some suggestions: principal component analysis (PCA) should be included. How did you check the quality of RNA, what were the criteria? How did you perform downstream analysis? Please write more details about the evaluation of transcriptome data, definition of gene categories etc.

As suggested, we revised this section and provide detailed methods for RNA-seq analysis. We also include principal component analysis (PCA) in the revised manuscript.

Page 17, lines 363-365 Figure S6

–Discussion: Generally, I suggest that Authors rewrite and/or reedit the discussion because it many times repeats the previously described facts and observations in Results without further detailed explanations/hypothesis/extrapolation etc. Moreover, it has crucial importance that the observed results should be translated into clinical aspect regarding *C. auris* infection.

As suggested, we revised the Discussion section and include the implications.

REVIEWERS' COMMENTS

Reviewer #1 (Remarks to the Author):

The revised manuscript by Bing et al describes studies stemming from the observation that in vivo passage of some *C. auris* strains contributes to the formation of adapted derivative strains which exhibit an aggregative phenotype with unique growth and in vivo fitness characteristics. The authors have responded to many reviewer comments and concerns and considerably revised the manuscript, including the performance of a number of additional studies. Generally, the majority of major concerns have been addressed. Only a small number of moderate and minor concerns remain and should be addressed prior to publication:

Moderate concerns-

Lines 330 to 331 and line 672: stating that mutations in regulators of budding or division are "common among *C. auris* clinical isolates" seems overstated/ under supported by the finding that when assessing 31 genes across over 4000 isolates of *C. auris*, 11 instances of a mutation (~0.2% of isolates assessed) were noted. Please revise.

Throughout- Phrasing should be carefully chosen when assuming the function of uncharacterized *C. auris* genes based on homology with genes characterized in other yeast. Such as (but not limited to) lines 302 to 305 where the 10 genes referred to have not all been shown to regulate CWI in *C. auris* but are predicted/assumed to based on homology.

Lines 592 to 595: it is not clear the difference in virulence/ in host fitness is caused by the model used, it is certainly possible, but the use of "likely" seems inappropriate as other explanations exist.

Lines 694 to 697: The prevalence of aggregating phenotypes such as those described in this manuscript has not been assessed across a large collection of *C. auris* clinical isolates so referring to the phenotype as having a "wide prevalence" seems unjustified at this time. This is particularly true if the authors are specifically referring to the aggregation caused by failure to separate rather than increased adhesion.

Minor concerns-

Lines 169 to 172: The manuscript cited does support the idea that a subset of *C. auris* isolates can form aggregates in vivo, but to say that "most clinical isolates are capable of forming aggregates in the host" seems like an over statement considering none of the Clade I or Clade IV isolates in this study were observed to form aggregates.

Line 318: Figure S4 does not show the evolved strains in the BJCA001 background but rather shows the strains constructed by targeted genetic manipulation.

Lines 380 to 381: The cause of the smaller number of DEGs associated with CHS1 is not known, phrasing should be changed to make clear the authors hypothesize/ suggest it might be related to the step/ position of CHS1 in the CWI pathway.

Reviewer #2 (Remarks to the Author):

Although there remains some issues with the way the data are described, I continue to believe this is an important and significant study that is worthy of high profile. But in my view further revision (relatively minor) is required as described below.

Rapid evolution of an adaptive multicellular morphology of *Candida auris* during systemic infection in the host. Nature Communications

Overall this is an interesting and exciting paper. The authors have improved the manuscript in the revision, but there remains some aspects of the way the narrative is presented that need to be rectified before acceptance.

Abstract: "Interestingly, LL-37 and PACAP are in turn able to induce the formation of the aggregative phenotype by causing genetic mutations under in vitro culture conditions." I still do not agree that this is mutagenesis. You could say this occurs by the by selection of spontaneous mutations that have a fitness advantage under in vivo culture conditions.

The same point applies for the section beginning on line 530. "Host antimicrobial peptides LL-37 and PACAP induce the aggregative morphology through genetic mutations". / as an alternative .../ .Genetic mutations resulting in the aggregative morphology had a fitness advantage in the presence of the host antimicrobial peptides LL-37 and PACAP.

If they believe that these peptide induce mutations directly they need to demonstrate that for example in a change in the measurable mutation frequency using a reporter system. I do not think this is what they actually wish to infer – rather that spontaneous mutants with an aggregating phenotype have a competitive advantage when they arise in vivo.

Line 83 and ears of immunocompetent individuals, / and skin 83 immunocompetent individuals,

Line 223. "suggesting that this host-induced mutation is a general feature of clinical strains." Again There is no direct evidence that the host-induces mutations, rather spontaneous mutations are enriched for in the in vivo environment and outgrow the non-mutated cells lacking this fitness advantage.

Line 230-31. "Our results indicate that the mammalian host is able to induce the formation of this multicellular aggregative morphology during systemic infections." / Suggest in stead....Our results indicate that the mammalian host provides a selective advantage for for cells capable of generating the multicellular aggregative morphology during systemic infections.

In Table S3 For MNN11 and VAN1 cite "Dean N, Jones R, DaSilva J, Chionchio G, Ng H. The Mnn10/Anp1-dependent N-linked outer chain glycan is dispensable for *Candida albicans* cell wall integrity. *Genetics*. 2022 May 5;221(1):iyac048. doi: 10.1093/genetics/iyac048. PMID: 35333306; PMCID: PMC9071539."

Noting that OCH1 is another gene that comes out of their screen they might note that they have 3 hits for the MPol complex for N-mannosylation (OC1, MNN11 And VAN1).

Line 5903. They should note that ace2 mutants of *C. glabrata* were found to be aggregated and the mutant was hyper-virulen and cite. Kamran M, Calcagno AM, Findon H, Bignell E, Jones MD, Warn P, Hopkins P, Denning DW, Butler G, Rogers T, Mühlischlegel FA, Haynes K. Inactivation of transcription factor gene ACE2 in the fungal pathogen *Candida glabrata* results in hypervirulence. *Eukaryot Cell*. 2004 Apr;3(2):546-52. doi: 10.1128/EC.3.2.546-552.2004. PMID: 15075283; PMCID: PMC387657.

Minor edits:

Line 719. (weighting 16-18 g)/ (weighing 16-18 g)

Line 813 five/5 – for consistency – also lines 864, 905, 915, 1236, 1255, 1260etc etc

Note: hours/ h, u/ μ (e.g. line 903)

Some sentences in methods start with a numeral (e.g. lines 903, 907)

Reviewer #3 (Remarks to the Author):

The performed changes significantly improved the manuscript. I have no further comment.

Response to reviewers' comments

Our point-to-point response to reviewers' comments is highlighted in blue.

Reviewer #1 (Remarks to the Author):

The revised manuscript by Bing et al describes studies stemming from the observation that in vivo passage of some *C. auris* strains contributes to the formation of adapted derivative strains which exhibit an aggregative phenotype with unique growth and in vivo fitness characteristics. The authors have responded to many reviewer comments and concerns and considerably revised the manuscript, including the performance of a number of additional studies. Generally, the majority of major concerns have been addressed. Only a small number of moderate and minor concerns remain and should be addressed prior to publication:

Moderate concerns-

Lines 330 to 331 and line 672: stating that mutations in regulators of budding or division are "common among *C. auris* clinical isolates" seems overstated/ under supported by the finding that when assessing 31 genes across over 4000 isolates of *C. auris*, 11 instances of a mutation (~0.2% of isolates assessed) were noted. Please revise.

Revised.

Now stated as "These findings indicate that mutations in cell budding- or cell division-associated regulators are present in *C. auris* clinical strains." Page 14 lines 319-320
"suggesting that cellular aggregation caused by genetic mutations can be found in *C. auris* clinical isolates." Page 30 lines 665

Throughout- Phrasing should be carefully chosen when assuming the function of uncharacterized *C. auris* genes based on homology with genes characterized in other yeast. Such as (but not limited to) lines 302 to 305 where the 10 genes referred to have not all been shown to regulate CWI in *C. auris* but are predicted/assumed to based on homology.

Thanks for pointing this out. All associated genes are included in Figure 3b and 3c. To clarify this point, we added a sentence to the figure legend.

"Note that the biological processes or signaling pathways of the *C. auris* mutated genes shown were predicted based on homology to their *S. cerevisiae* or *C. albicans* counterparts (panels b and c)." page 59 lines 1291-1293

Lines 592 to 595: it is not clear the difference in virulence/ in host fitness is caused by the model used, it is certainly possible, but the use of "likely" seems inappropriate as other explanations exist.

Revised.

"...could be due to the different strains and infection models used."

Page 27 lines 586-587

Lines 694 to 697: The prevalence of aggregating phenotypes such as those described in this

manuscript has not been assessed across a large collection of *C. auris* clinical isolates so referring to the phenotype as having a "wide prevalence" seems unjustified at this time. This is particularly true if the authors are specifically referring to the aggregation caused by failure to separate rather than increased adhesion.

Revised.

Page 31 lines 688-690

Minor concerns-

Lines 169 to 172: The manuscript cited does support the idea that a subset of *C. auris* isolates can form aggregates in vivo, but to say that "most clinical isolates are capable of forming aggregates in the host" seems like an over statement considering none of the Clade I or Clade IV isolates in this study were observed to form aggregates.

Revised. The literature does support the claim that numerous clinical isolates can form aggregates. In terms of the host, we edited this to say "suggesting that numerous" We edited the wording for accuracy throughout.

Line 318: Figure S4 does not show the evolved strains in the BJCA001 background but rather shows the strains constructed by targeted genetic manipulation.

Revised.

"...in the control strain."

Lines 380 to 381: The cause of the smaller number of DEGs associated with CHS1 is not known, phrasing should be changed to make clear the authors hypothesize/ suggest it might be related to the step/ position of CHS1 in the CWI pathway.

Revised. Page 17 lines 371-372

Reviewer #2 (Remarks to the Author):

Although there remains some issues with the way the data are described, I continue to believe this is an important and significant study that is worthy of high profile. But in my view further revision (relatively minor) is required as described below.

Rapid evolution of an adaptive multicellular morphology of *Candida auris* during systemic infection in the host. Nature Communications

Overall this is an interesting and exciting paper. The authors have improved the manuscript in the revision, but there remains some aspects of the way the narrative is presented that need to be rectified before acceptance.

Abstract: "Interestingly, LL-37 and PACAP are in turn able to induce the formation of the aggregative phenotype by causing genetic mutations under in vitro culture conditions." I still do not agree that this is mutagenesis. You could say this occurs by the by selection of spontaneous mutations that have a fitness advantage under in vivo culture conditions.

We revised the Abstract section based on the reviewer's suggestion.

The same point applies for the section beginning on line 530. “Host antimicrobial peptides LL-37 and PACAP induce the aggregative morphology through genetic mutations”. / as an alternative .../. Genetic mutations resulting in the aggregative morphology had a fitness advantage in the presence of the host antimicrobial peptides LL-37 and PACAP.

We revised the subtitle for accuracy based on the content of the section and mention the fitness advantage in the text.

If they believe that these peptide induce mutations directly they need to demonstrate that for example in a change in the measurable mutation frequency using a reporter system. I do not think this is what they actually wish to infer – rather that spontaneous mutants with an aggregating phenotype have a competitive advantage when they arise in vivo.

Agreed – we do not wish to infer this. This point should now be clear in the revised text.

Line 83 and ears of immunocompetent individuals, / and skin 83 immunocompetent individuals,
Revised.

Line 223. “suggesting that this host-induced mutation is a general feature of clinical strains.” Again There is no direct evidence that the host-induces mutations, rather spontaneous mutations are enriched for in the in vivo environment and outgrow the non-mutated cells lacking this fitness advantage.

Removed this statement.

Line 230-31. “Our results indicate that the mammalian host is able to induce the formation of this multicellular aggregative morphology during systemic infections.” / Suggest instead....Our results indicate that the mammalian host provides a selective advantage for for cells capable of generating the multicellular aggregative morphology during systemic infections.

Revised as suggested. Page 10 lines 216-218

In Table S3 For MNN11 and VAN1 cite “Dean N, Jones R, DaSilva J, Chionchio G, Ng H. The Mnn10/Anp1-dependent N-linked outer chain glycan is dispensable for *Candida albicans* cell wall integrity. *Genetics*. 2022 May 5;221(1):iyac048. doi: 10.1093/genetics/iyac048. PMID: 35333306; PMCID: PMC9071539.”

Cited.

Noting that OCH1 is another gene that comes out of their screen they might note that they have 3 hits for the MPol complex for N-mannosylation (OC1, MNN11 And VAN1).

Mentioned in the revised text. Page 13 lines 292-294

Line 5903. They should note that ace2 mutants of *C. glabrata* were found to be aggregated and the mutant was hyper-virulent and cite. Kamran M, Calcagno AM, Findon H, Bignell E, Jones MD, Warn P, Hopkins P, Denning DW, Butler G, Rogers T, Mühlischlegel FA, Haynes K. Inactivation of transcription factor gene ACE2 in the fungal pathogen *Candida glabrata* results in hypervirulence. *Eukaryot Cell*. 2004 Apr;3(2):546-52. doi: 10.1128/EC.3.2.546-552.2004. PMID: 15075283; PMCID:

PMC387657.

Cited. Page 12 line 263

Minor edits:

Line 719. (weighting 16-18 g)/ (weighing 16-18 g)

Revised.

Line 813 five/5 – for consistency – also lines 864, 905, 915, 1236, 1255, 1260etc etc

Revised.

Note: hours/ h, u/ μ (e.g. line 903)

Revised.

Some sentences in methods start with a numeral (e.g. lines 903, 907)

Revised.

Reviewer #3 (Remarks to the Author):

The performed changes significantly improved the manuscript. I have no further comment.

Thanks!